# HEXIM1 inter-monomer autoinhibition governs 7SK RNA binding specificity and P-TEFb inactivation

Yuan Yang [1] ✉, Maria Grazia Murrali[1], Sabrina Galvan[1], Yaqiang Wang [1,2], Christine Stephen[1], Neha Ajjampore[1], Xiaoyu Wang[1] & Juli Feigon [1] ✉

Hexim proteins are key RNA-dependent regulators of eukaryotic transcription through 7SK-dependent sequestration and inactivation of the kinase P-TEFb (Cdk9–CyclinT1/2) in the 7SK RNP. P-TEFb activity drives release of RNA polymerase II from promoter-proximal pausing for eukaryotic and HIV-1 transcription. The molecular mechanism by which 7SK binding overcomes an intrinsic Hexim autoinhibition for subsequent P-TEFb inactivation has remained unresolved. Here, using NMR and biophysical methods we demonstrate that Hexim1 homodimer engages two high-affinity sites on 7SK RNA. This dual-site binding triggers a conformational rearrangement in Hexim1's disordered central region that unmasks the Cdk9-binding site, which is otherwise sequestered within an inter-monomer dimer interface. These findings reveal how Hexim autoinhibition dictates its specificity for 7SK RNA and prevents premature P-TEFb inhibition in the absence of 7SK, thereby providing a mechanistic understanding of Hexim/P-TEFb assembly into the 7SK RNP and further considerations for understanding Hexim–Tat competition during viral transcription.

HEXIM (HEXamethylene-bis-acetamide-Inducible protein in vascular smooth Muscle cells)[1] is a unique family of RNA-dependent regulatory proteins[2] whose principal target is 7SK RNA, an abundant ~332-nucleotide non-coding RNA that plays a central role in regulating RNA polymerase II (Pol II) transcription. In addition to 7SK, Hexim has been shown to interact with other RNAs, including NEAT1[3] and several mRNAs[4], hinting at broader cellular functions. Mammals express two paralogs of Hexim (Hexim1 and Hexim2)which differ in tissue distribution and can functionally compensate for one another within the 7SK ribonucleoprotein complex (7SK RNP)[5,6]. Hexim suppresses the kinase activity of positive transcription elongation factor b (P-TEFb) by sequestering it within the 7SK RNP. Crucially, this sequestration depends on 7SK RNA binding to Hexim[7–9]. P-TEFb is a heterodimer comprising the catalytic subunit Cdk9 and a regulatory subunit, Cyclin T1 or T2 (CycT1/CycT2), which exhibits specificity for Hexim1 and Hexim2, respectively[10]. Treatment of live cells with a Cdk9 inhibitor

mobilizes hnRNPs to bind to 7SK and consequently evict both Hexim and P-TEFb, not associated with each other, from 7SK within minutes[11]. Upon release from 7SK RNP, P-TEFb phosphorylates the C-terminal domain (CTD) of Pol II along with several other transcription factors (NELF, DSIF, PAF, SPT6), a process essential for Pol II to transition from promoter-proximal paused to productive elongation[12,13]. Beyond its host regulatory roles, 7SK RNP is also a critical target for HIV-1 replication, where Tat (Trans-activator of transcription) protein hijacks the 7SK RNP to the HIV-1 promoter and extracts P-TEFb into a viral super elongation complex via the nascent TAR (trans-activation response) RNA element[14–17].

Hexim1 contains three main regions: (i) the N-terminal region (1-149), which is poorly conserved across vertebrates (Supplementary Figs. 1 and 2) and predicted to be intrinsically disordered (Supplementary Fig. 3), (ii) the central region (150-253), discussed below, and (iii) a C-terminal coiled-coil domain (CC hereafter, 254-359) that has

[1]Department of Chemistry and Biochemistry, University of California Los Angeles, Los Angeles, CA, USA. [2]Present address: Departments of Biophysics and Obstetrics & Gynecology, Medical College of Wisconsin, Milwaukee, WI, USA. ✉e-mail: yuanyangwhu@ucla.edu; feigon@mbi.ucla.edu

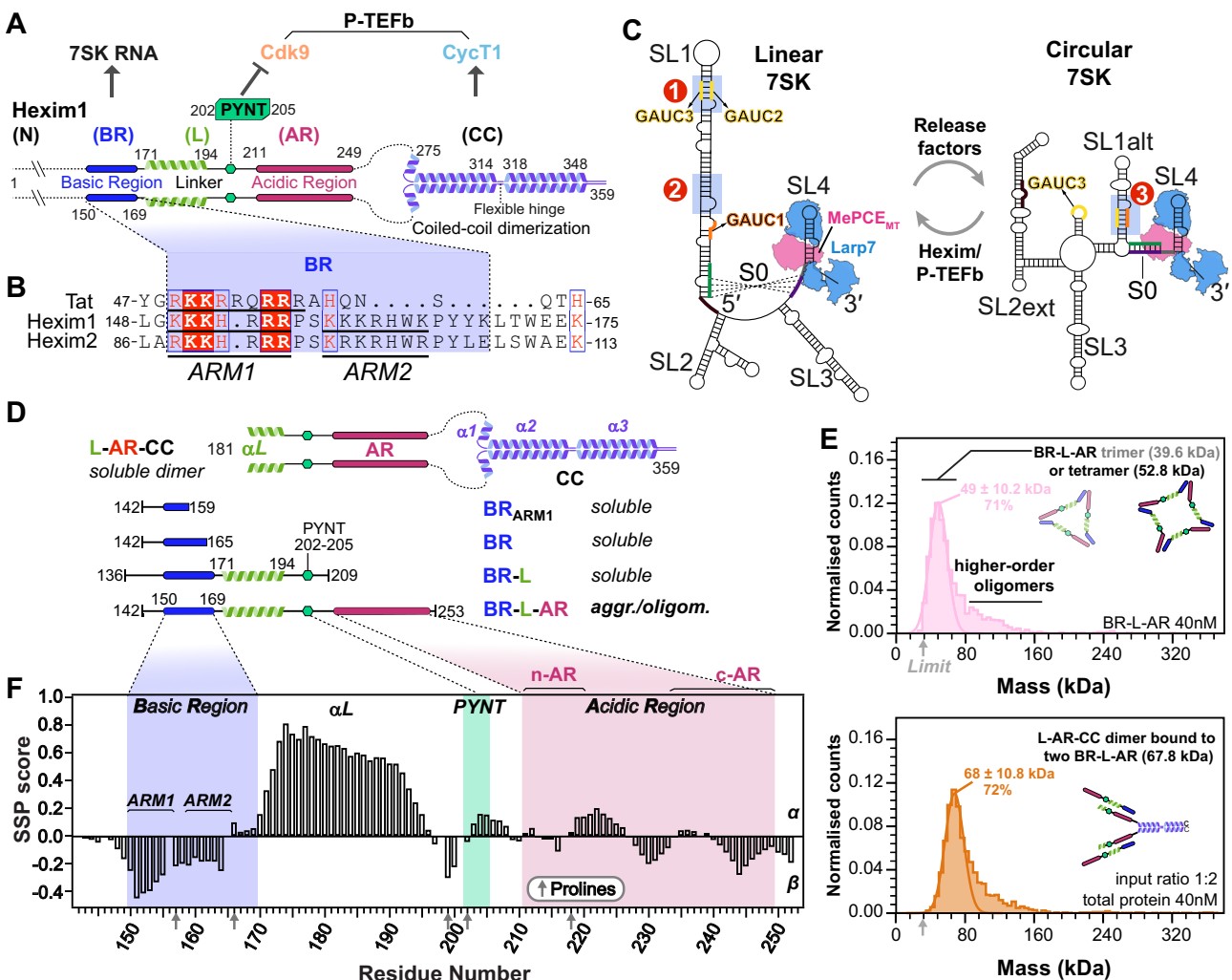

**Fig. 1 | Domain schematics and secondary structures of Hexim1 and 7SK core RNPs in two RNA conformations that affect available Hexim binding sites.**
**A** Domain schematic of Hexim1 homodimer, with interacting partners of Hexim1 structural motifs indicated on top. The structural segments of Hexim1 are labeled as N-terminus (N), Basic Region (BR), Linker (L), Acidic Region (AR), and coiled-coil domain (CC). **B** Sequence alignment between HIV-1 Tat (accession ID P04608, group M subtype B), Hexim1 and Hexim2 RNA binding domain.
**C** MePCE−7SK−Larp7 core RNPs in linear and circular conformations. Hexim binding sites investigated in this study are labeled (Site1, Site2, Site3). MePCE (magenta) and Larp7 (blue) are illustrated based on cryo-EM structures[34].
**D** Schematics of Hexim1 constructs. Soluble or aggr./oligom. (aggregated/ oligomerized) indicate behavior in solution. **E** Mass photometry data showing formation of intermolecular complexes for BR-L-AR (top) and a 1:2 mixture of L-AR-CC and BR-L-AR (bottom). The observed Mass Photometry peak corresponds to a molecular weight of $49 \pm 10.2$ kDa−between the theoretical trimer (39.6 kDa) and tetramer (52.8 kDa) sizes, but closer to the latter. This shift likely results from the trimer peak being near the 30 kDa detection limit (gray arrow), causing partial peak truncation and overlap with the tetramer peak. Monomer and dimer species fall below the detection threshold and are not observed. **F** Secondary structure propensity (SSP) scores for BR-L-AR, plotted as a function of residue number. Gray arrows on the X-axis indicate positions of proline residues, which lack amide protons, resulting in the absence of SSP values for the residues preceding prolines.

been shown by NMR to form a stable homodimer[18] and mediates Hexim1 binding to CycT1[19] (Fig. 1A). Within the central region only the Basic Region (BR) has been structurally characterized[7,8]. BR directly interacts with RNA and comprises two Arg/Lys-rich stretches separated by a conserved proline-serine and is often referred to as ARM (arginine-rich motif)[7,20] due to its sequence similarity to the single ARM of HIV-1 Tat (Fig. 1B). Following the BR is a linker containing a conserved PYNT motif, known to bind Cdk9[7,21] and act in concert with the CC to inactivate P-TEFb[9] (Fig. 1A). The linker is followed by an Acidic Region (AR). A prevailing model proposes that an internal electrostatic interaction between the BR and AR maintains Hexim in an autoinhibited state that cannot bind P-TEFb, rendering P-TEFb inhibition RNA-dependent[9,22]. However, the molecular mechanism by which this autoinhibition prevents PYNT−Cdk9 engagement in the absence of RNA remains unknown.

Hexim−RNA binding specificity and stoichiometry have long been subjects of debate. One study suggested Hexim1 acts as a general binder of double-stranded RNA[23], whereas structural studies[24–27] have focused on a specific motif−a GAUC palindrome−within stem-loop 1 (SL1) of 7SK RNA. This site (Site1 in Fig. 1C), encompassing GAUC2 and GAUC3 motifs (nts 42-45 and 64-67), is a conserved marker for 7SK genes[28,29] and also serves as the RNA-binding site for HIV-1 Tat[26,30]. However, a biochemical study[30] identified a second Hexim1 binding site in SL1 lacking a GAUC motif (Site2 in Fig. 1C). Lastly, another GAUC motif (GAUC1, nts 13-16) is present in proximal SL1 within one RNA helical turn of the binding/capping site for methylphosphate capping enzyme (MePCE)[31].

The issues of Hexim binding specificity and stoichiometry are further complicated by evidence[32,33], most recently from structural studies[34] and RNA chemical probing by DANCE-MaP[35], of at least two

distinct 7SK RNA conformations in vitro and in cells: a linear form, where a long SL1 presents the Hexim binding sites discussed above, and a circular form, in which a shorter SL1alt forms a rearranged GAUC palindrome (GAUC1 and GAUC2; Site3 in Fig. 1C). Two core protein components, MePCE and Larp7, constitutively assemble with 7SK RNA[36] for both linear and circular conformations, as shown by cryo-EM structures[34] (Fig. 1C). Notably, switching between P-TEFb-bound and -free 7SK RNP pools, likely corresponding to the two RNA conformations, is required for stress-induced regulation of P-TEFb release[34,35,37].

In this study, we used biophysical methods to characterize the central region of Hexim1, comprehensively map Hexim1 binding sites on 7SK RNA, and determine the binding stoichiometry. Paramagnetic relaxation enhancement (PRE) NMR measurements reveal that Hexim1 autoinhibition is governed by inter-monomer interaction, where the PYNT motif contacts both BR and AR regions. RNA binding to BR disrupts this interaction, unmasking the PYNT site. Complementary NMR and isothermal titration calorimetry (ITC) experiments further demonstrate that these inter-monomer interactions enhance binding specificity for linear 7SK over the circular conformation. This specificity is achieved through a cooperative recognition of dual high-affinity sites in SL1. Importantly, the sequestration of BR in the autoinhibited state weakens affinity for individual RNA sites, highlighting that both high-affinity sites are required to fully release Hexim1 autoinhibition. Together, these findings uncover an intricate network of intra-dimer interaction in Hexim1−driven by intrinsically disordered regions−that fine-tunes its RNA-binding behavior. Our results explain how RNA-mediated release of Hexim1 autoinhibition governs P-TEFb binding and inactivation, and how 7SK conformational switching regulates the partitioning of Hexim/P-TEFb in 7SK RNP.

## Results

### Intrinsically disordered Hexim1 BR-L-AR is prone to inter-molecular interaction

To characterize the Hexim1 central region, we expressed and purified a monomeric construct (residues 142–253) encompassing BR, linker (L) containing PYNT motif, and AR (hereafter referred to as BR-L-AR). Several additionally truncated Hexim1 monomeric constructs, BR$_{ARM1}$, BR and BR-L (Fig. 1D), were used to compare with BR-L-AR. We observed a solubility limit of ~70 μM for BR-L-AR despite testing many conditions. In contrast, BR$_{ARM1}$, BR, and BR-L are easily soluble at ~0.5−0.8 mM under similar conditions. This suggests that BR-L-AR is prone to aggregation, mediated by intermolecular interactions that require both BR and AR. To detect the species formed via inter-molecular interactions, we used native polyacrylamide gel electrophoresis (NativePAGE[38]) and mass photometry[39,40] at ~4−25 μM and 40 nM, respectively. A clear ladder pattern of monomer and various oligomers was observed in NativePAGE for BR-L-AR, and mass photometry data captured trimer and tetramer species (Supplementary Fig. 4A−C and Fig. 1E). This indicates that the intermolecular interaction of BR-L-AR is stable enough to be observed even at 40 nM.

In contrast, freshly purified Hexim1 and L-AR-CC dimeric proteins both migrated as a single major band in NativePAGE, and mass photometry showed a homogeneous particle distribution corresponding to dimers (Fig. 1A, D and Supplementary Fig. 4A, D, E). However, when we combined BR-L-AR with either dimeric protein, we observed a stable intermolecular complex by mass photometry (Fig. 1E and Supplementary Fig. 4D, E). Together, these data suggest that in full-length Hexim1, the BR-L-AR from each monomer interacts with each other, consistent with the previous model that interaction between BR and AR maintains Hexim in an autoinhibited state[9,22]. These interactions are robust enough to prevent oligomerization of the dimeric Hexim1, but are readily competed by adding BR-L-AR monomeric protein in trans, suggesting relatively fast exchange dynamics, consistent with the nature of interactions formed by charged disordered protein regions (IDR)[41].

To determine the extent of disorder and secondary structure in BR-L-AR, we used multi-dimensional heteronuclear NMR experiments on uniformly $^{15}$N,$^{13}$C-labeled protein (see "Methods"). Despite the low BR-L-AR concentration (<70 μM) we could achieve due to inter-molecular interactions, we were able to assign $^1H_N$, $^{15}$N, $^1H\alpha$, $^{13}C\alpha$, and $^{13}C\beta$ chemical shifts and calculate secondary structure propensity (SSP) scores[42] using Cα and Cβ chemical shifts (Fig. 1F and Supplementary Fig. 5A). The SSP scores, ranging from +1 to −1, reflect the fraction of α- or β- structure formed by each residue. From the SSP scores, we identified an α-helix (aa 171–194) in the linker region, between BR and PYNT motif (Fig. 1F). We also defined the two Arg/Lys-rich regions in BR as ARM1 and ARM2 (Fig. 1C): of these, ARM1 (residues 151–154) shows a moderate propensity to form β-structure, with a minimum SSP reaching −0.45. The rest of BR-L-AR is largely disordered, with SSP scores within +0.3 to −0.3 (Fig. 1F).

In summary, BR-L-AR is largely intrinsically disordered, contains a short α-helix in the linker following the BR, and has a strong tendency to form intermolecular interactions with other monomers and with Hexim1 dimer.

### Characterization of Hexim monomer−7SK RNA Site1 interaction

To look at the interaction of BR-L-AR with the previously characterized single 7SK RNA binding site (SL1 Site1 in Fig. 1C)[24−27,43,44], we first investigated the structure of two designed RNA constructs from the distal end of linear 7SK SL1 (SL1-dI and SL1-dI$_{\Delta U}$) using NMR (Fig. 2A, C). We assigned N-H⋯N imino (base-paired U/G) and non-exchangeable base and H1′ proton (all nucleotides) resonances for SL1-dI (Supplementary Figs. 6−8). Our analysis confirmed its complex architecture, which consists of a base-paired stem flanked by two dynamic U-rich bulges, consistent with previous structural studies[24−27,43,44] (Fig. 2A). Those studies had variously reported that the two flanking U-rich bulges (U$_{40}$U$_{41}$ and U$_{63}$) can form base triples with the A$_{43}$-U$_{66}$ and U$_{44}$-A$_{65}$ or G$_{42}$-C$_{67}$ base pairs, respectively; however, which of these base triples were observed varied with sample conditions including pH[25,26,43,44]. For SL1-dI, at 150 mM KCl and pH 6.2, we observed a stable U$_{40}$⋯A$_{43}$-U$_{66}$ base-triple and a dynamic U$_{63}$⋯A$_{65}$-U$_{44}$, where the U$_{63}$ bulge samples at least two states, likely between stacking on G$_{64}$ and Hoogsteen paring with A$_{65}$ (Fig. 2A and Supplementary Fig. 6A, C). To investigate the relative dynamics of these nucleotides, the non-exchangeable resonance intensities from non-constant time $^{13}$C-$^1$H HQSC spectra were acquired, and normalized $^{13}$C intensities were plotted, where higher normalized intensity generally reflects faster dynamics on the pico-to-nanosecond timescale[45] (Fig. 2A). These measurements confirmed that U$_{40}$ is stable (low intensities of 0.1 as helical stem) while U$_{41}$, U$_{63}$, and A$_{65}$ are much more flexible (elevated intensities) consistent with our observation of one stable and one dynamic base triple (Fig. 2A). At pH 5.2, both these base triples were stably observed[26].

Next, we titrated BR-L-AR into the RNAs and monitored chemical shift perturbations (CSP). On the RNA side, for SL1-dI, the entire GAUC2-GAUC3 stem, U$_{40}$U$_{41}$ and U$_{63}$ bulges are significantly shifted, with G$_{64}$ affected the most (Fig. 2B, C and Supplementary Fig. 6). For SL1-dI$_{\Delta U}$, although the same overall Site1 was used for binding, the distal half of GAUC2-GAUC3 stem exhibits smaller CSP (Fig. 2C and Supplementary Fig. 8), indicating a moderate contribution of U$_{63}$ bulge (deleted in SL1-dI$_{\Delta U}$) to Hexim1 binding, consistent with an EMSA binding assay[44].

On the protein side, $^{15}$N-$^1$H HSQC spectra of BR-L-AR titrated with either SL1-dI$_{\Delta U}$ or SL1-dI showed significant chemical shift changes and severe peak broadening for protein residues 151-170 spanning the entire BR (including ARM1, ARM2, and surrounding sequence) (Fig. 2D, E and Supplementary Fig. 9). Less line broadening is observed for BR-L-AR when SL1-dI$_{\Delta U}$ is added than for SL1-dI, indicative of a shift towards fast exchange on the NMR timescale likely due to a faster off-rate[46]. This allowed better tracing of the peak migration trajectories and observation of the BR residue S158 in the RNA-bound spectrum to

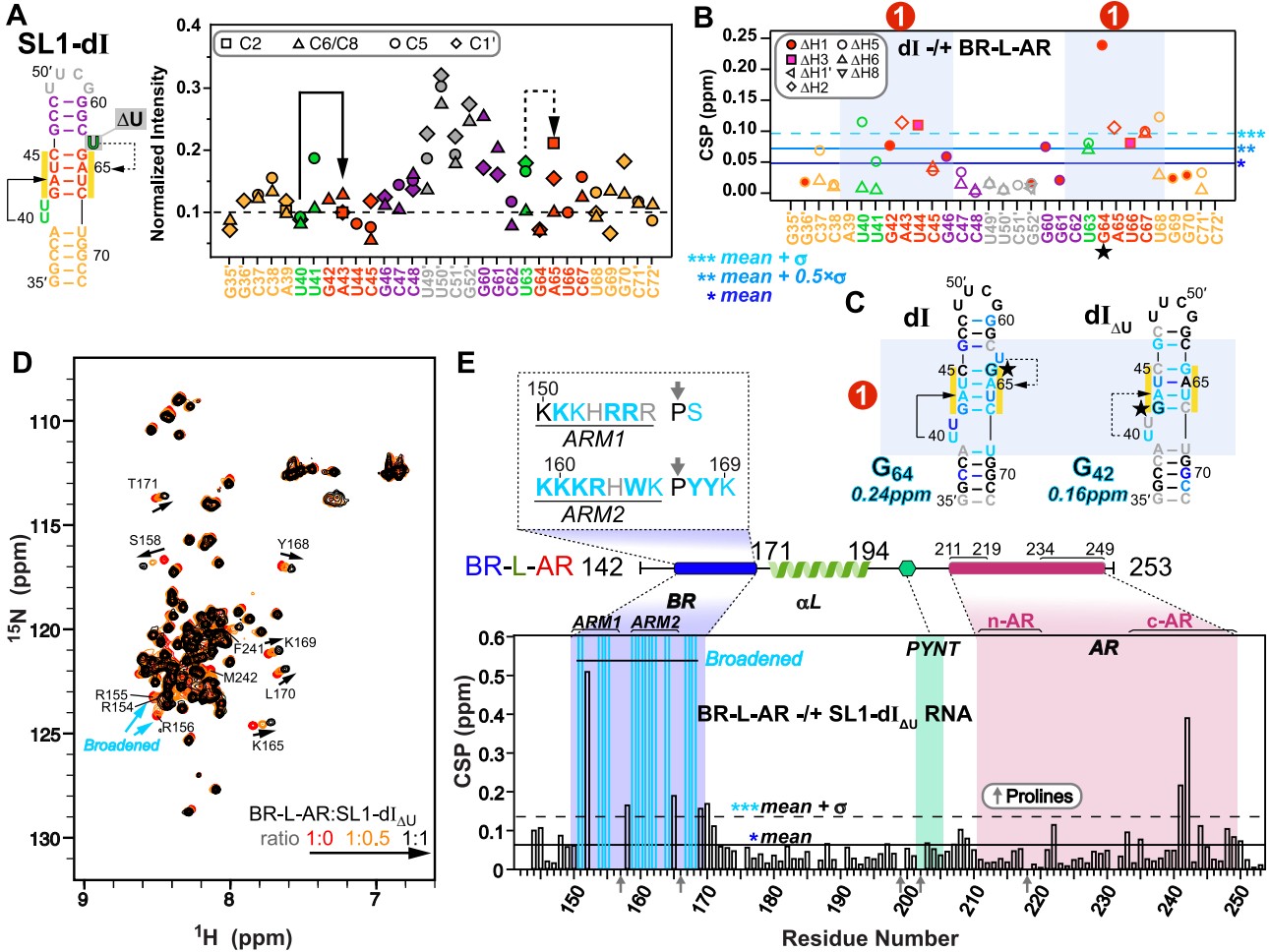

**Fig. 2 | NMR characterization of Hexim1 BR-L-AR–Site1 interaction.**
**A** Normalized $^{13}$C intensities of SL1-dI (schematic at left) from non-constant-time $^{13}$C-$^1$H HSQC spectra. The intensities of each C–H spin type were normalized against the lowest intensity from the RNA helical stem to the reference value of 0.1. The nucleotides are colored by subdomain as shown in the schematic at left. **B** Chemical shift perturbations (CSP) plot of SL1-dI base and H1' proton resonances upon binding BR-L-AR. **C** Secondary structure of Site1 RNAs (SL1-dI and SL1-dI$_{\Delta U}$) colored by CSP upon binding BR-L-AR. Nucleotide with the largest CSP is highlighted with a black/cyan outline and a star, with their CSP value indicated at the bottom. **D** 500 MHz $^{15}$N-$^1$H HSQC spectra of BR-L-AR (red) with addition of linear 7SK SL1-

dI$_{\Delta U}$ at 1:0.5 (orange) and 1:1 (black). Representative resonances that shift significantly or broaden upon RNA binding are labeled with residue assignments; black arrows indicate direction of shift. **E** CSP of BR-L-AR upon binding 7SK SL1-dI$_{\Delta U}$ (1:1.5 ratio). Cyan bars indicate residues broadened beyond detection by RNA binding. Gray arrows on the X-axis in D and E indicate positions of proline residues, which lack amide protons, resulting in a lack of CSP values for the prolines. The dashed box insert shows color-coded RNA-binding induced CSPs on the BR amino acid sequence, where cyan bold residues indicate those with amide resonances broadened beyond detection upon RNA binding and cyan residues indicate those with significant chemical shift changes.

confirm the binding site (Fig. 2D, E). Interestingly, in both cases, the C-terminal region of AR (c-AR) exhibited chemical shift changes at residues Phe241-Met242. To verify that the c-AR shift is indirect, we monitored NMR titration of SL1-dI$_{\Delta U}$ into the shorter BR-L or BR and found that it bound the same way as BR-L-AR (Supplementary Fig. 10A–E). These results indicate 7SK RNA binding to BR induces an allosteric change in c-AR.

In summary, 7SK SL1 Site1 features a stem with two flanking bulges that can form one stable and one dynamic U···A-U base-triple, and both base-triples are required for specific BR binding that engages the entire GAUC-palindrome. Hexim1 uses both ARM1 and ARM2 of BR for RNA binding, and this binding induces a conformational change in BR-L-AR. In contrast to Hexim, HIV-1 Tat protein has only ARM1 that is necessary and sufficient for binding Site1 (Fig. 1C)[25,26,30], indicating a significant difference in Hexim and Tat recognition of 7SK RNA.

**Characterization of Hexim1 autoinhibited conformation by PRE**
To characterize the autoinhibitory interactions within the BR-L-AR region, we used Paramagnetic Relaxation Enhancement (PRE)[47] NMR.

This method can map weak and transient interactions[48,49] common in IDRs[50] by measuring distance-dependent line broadening from a covalently attached MTSL nitroxide spin label up to 25 Å away. A key challenge is distinguishing intra-monomer interactions from inter-monomer interactions. To dissect these, we established two distinct experimental setups using three individual cysteine substitution sites (S226C, S233C, S237C): (i) "total interaction" experiment: $^{15}$N-labeled and MTSL-tagged BR-L-AR, capturing both intra- and inter-monomer contacts (Fig. 3A, B and Supplementary Fig. 11); and (ii) "inter-only" experiment: 1:1 mixture of the reporter $^{15}$N-labeled, untagged BR-L-AR with the broadener $^{14}$N-labeled and MTSL-tagged BR-L-AR (Fig. 3C–E). In the latter, only inter-monomer interactions will result in line-broadening, however to a lesser extent than the "total" experiments, since only 50% of the "dimer" population is the combination of $^{15}$N and $^{14}$N-MTSL labeled proteins. To visualize and identify interactions between structural segments, we defined regions of primary broadening (red) and secondary broadening (orange) as stretches harboring 3 or more residues (either contiguous or non-contiguous) with intensity ratios lower

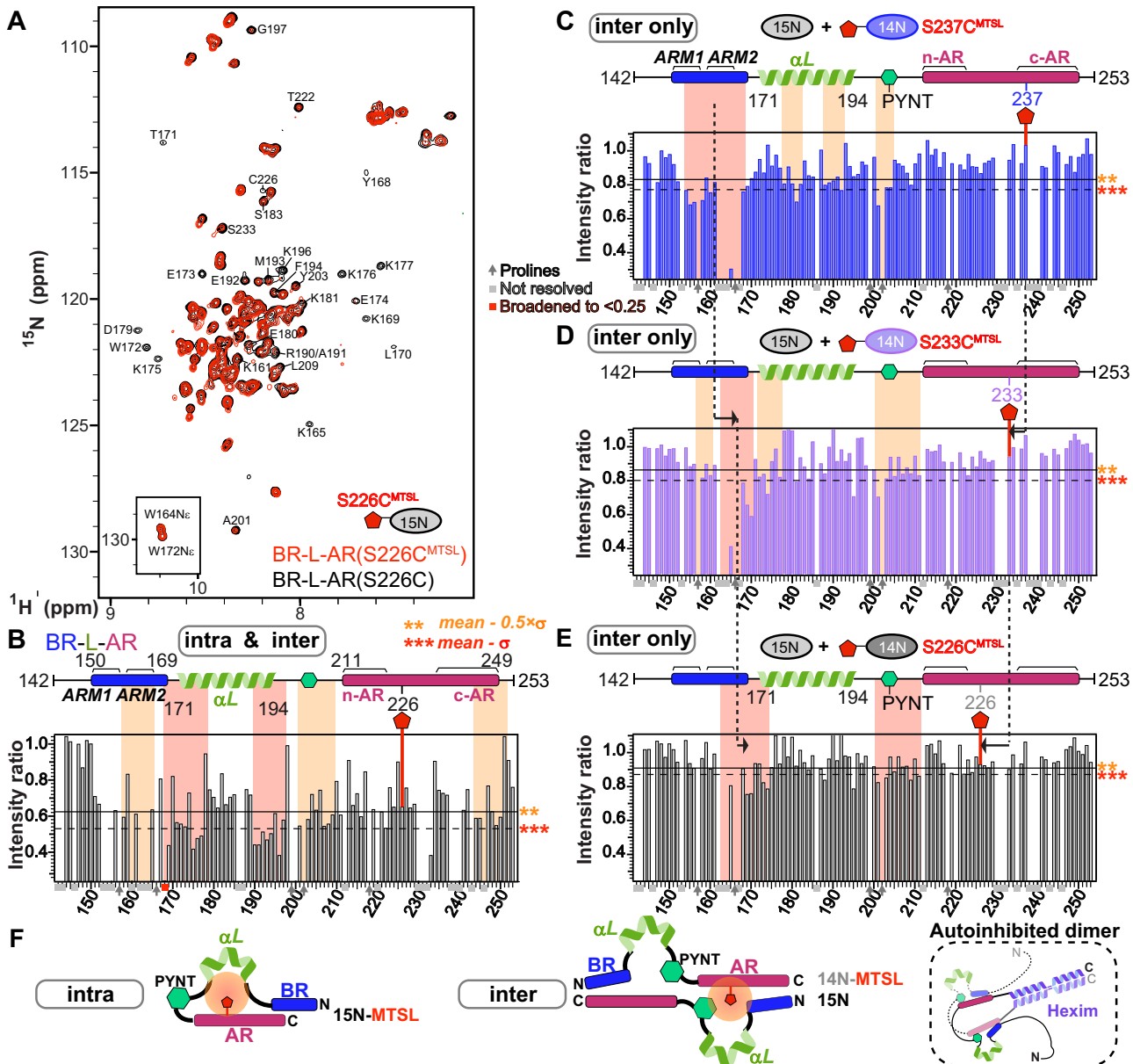

**Fig. 3 | Probing the intra- and inter-monomer components of Hexim1 auto-inhibited conformation. A** $^{15}$N-$^{1}$H HSQC spectral overlay of BR-L-AR(S226C) (black) and BR-L-AR(S226C$^{MTSL}$) (red). Residues that are significantly broadened by MTSL are labeled. **B** Paramagnetic Relaxation Enhancement (PRE) intensity ratios ($I_{PRE}/I_{noPRE}$) for BR-L-AR(S226C) with versus without MTSL tag of resolved resonances plotted against residue number. Schematic of BR-L-AR with S226C$^{MTSL}$ tag location is shown above plot. **C–E** Plots of intermolecular PRE intensity ratios of (**C**) $^{14}$N BR-L-AR(S237C$^{MTSL}$) (blue bars), (**D**) S233C$^{MTSL}$ (purple bars), and (**E**) S226C$^{MTSL}$ (dark gray bars) added to $^{15}$N BR-L-AR. Schematics of BR-L-AR and individual MTSL locations are shown above each plot. For (**B–E**) solid horizontal line indicates the value of [mean−0.5 × σ] and dashed line indicates the value of [mean−σ]. Stretches of > 3 either contiguous or non-contiguous residues with intensity ratios below the

dashed line [mean−σ] or solid line [mean−0.5 × σ] are highlighted with red and orange shaded boxes, respectively. Dark arrows denote proline residues, lacking amide proton, light gray boxes denote residues with overlapping resonances that are excluded for accurate intensity determination, and red box denote residues broadened below the lower Y-axis boundary. The dashed lines connecting (C-D-E) panels highlight the opposite direction of shifting of the primary broadening region relative to the directional change of the MTSL position in the protein sequence, indicating a head-to-tail conformation for the inter-molecular interactions.
**F** Simplified cartoons illustrating the intra- and inter-molecular components of the PRE detected interactions, and a self-hugging topology of autoinhibited Hexim1 dimer based on the inter-molecular interactions.

than [mean - σ] and lower than [mean - 0.5xσ], respectively (Fig. 3B–E).

In the inter-only experiments, we observed a distinct pattern: S237C$^{MTSL}$ primarily broadens the entire BR (154-168), with ARM2 broadened more severely than ARM1 (Fig. 3C); S233C$^{MTSL}$ broadens ARM2-PYYK (Fig. 3D); and S226C$^{MTSL}$ broadens ARM2-PYYK, beginning of αL, and PYNT motif (Fig. 3E). This pattern indicates an overall head-to-tail orientation for BR-L-AR inter-monomer interaction (Fig. 3F), where the primary broadened region (red) shifts in the N- to C-terminal

direction when we move the MTSL-tag from the C- to N-terminal direction (dashed lines in Fig. 3C–E).

We then analyzed the "total interaction" experiment by comparing this "total" profile to the "inter-only" S226C profile (compare Figs. 3B–E) to deconvolve the two interaction types. The secondary broadening (orange) observed in the "total" experiment at ARM2-PYYK and PYNT motif matched the "inter-only" experiment, confirming this as the inter-monomer component. Therefore, the remaining primary broadening, which was observed at both the beginning and the end of

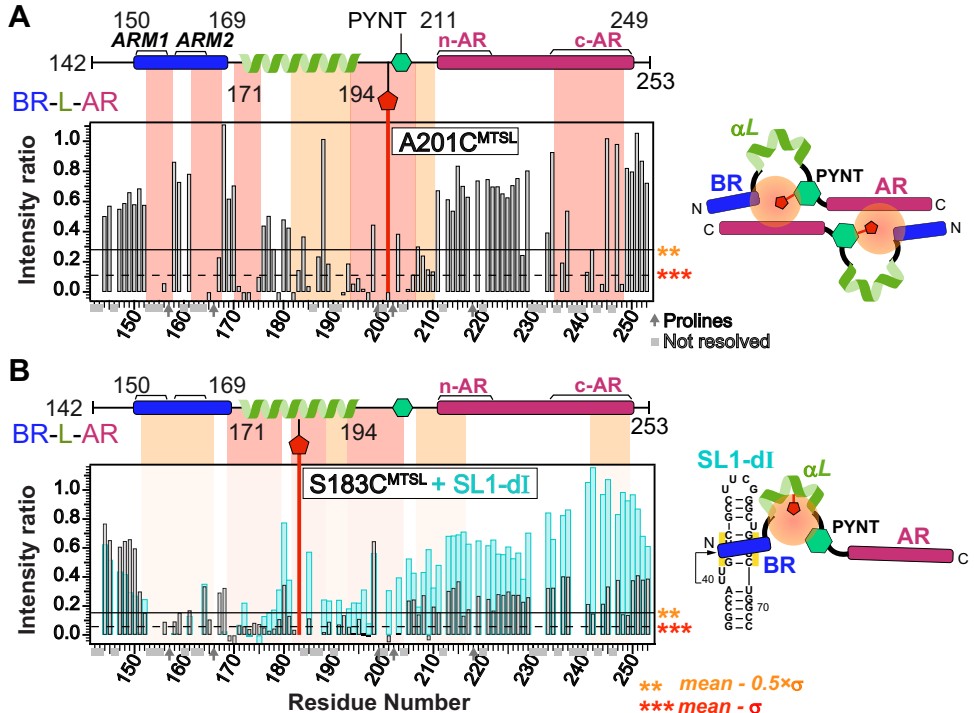

**Fig. 4 | PYNT[205] is the interaction hub for Hexim1 autoinhibited conformation and is unmasked upon 7SK RNA binding to BR. A** Schematic and PRE intensity ratios ($I_{PRE}/I_{noPRE}$) for $^{15}$N-labeled BR-L-AR(A201C) with versus without the MTSL tag of resolved resonances plotted against residue number. **B** Schematic and Paramagnetic Relaxation Enhancement (PRE) intensity ratios ($I_{PRE}/I_{noPRE}$) for $^{15}$N-labeled BR-L-AR(S183C) with versus without MTSL tag of resolved resonances plotted against residue number in the absence (black/gray bars) and presence (cyan bars) of SL1-dI RNA (1:1.2 protein:RNA ratio). For (**A**, **B**), the solid horizontal line indicates the value of [mean-0.5 × σ] and the dashed line indicates the value of [mean-σ].

Stretches of > 3 either contiguous or non-contiguous residues with intensity ratios below the dashed line [mean-σ] or solid line [mean-0.5 × σ] are highlighted with red and orange shaded boxes, respectively. Shaded boxes in panel B are from the PRE profile in the absence of RNA. Dark arrows denote proline residues, lacking amide proton, and light gray boxes denote residues with overlapping resonances that are excluded for accurate intensity determination. Residues with negative values reflect that the resonances were broadened to the noise level. Simplified cartoons on the right illustrate the BR–PYNT–AR inter-molecular interactions and how RNA binding results in the release of these interactions.

the αL, represents the intra-monomer component (Fig. 3B, F). This indicates that helix αL is bent or kinked, allowing its two ends to approach S226C. In the S237C$^{MTSL}$ "total" experiment, almost all the resonances are broadened more severely than S226C$^{MTSL}$ (Supplementary Fig. 11), indicating a significant role of c-AR in the autoinhibition, consistent with Phe241-Met242 exhibiting allosteric CSP upon RNA binding (Fig. 2E).

In summary, PRE NMR experiments distinguished and mapped intra- and inter-monomer interactions in Hexim1 BR-L-AR. Based on these results, the inter-monomer interaction adopts a head-to-tail orientation (N→C to C→N), which in full-length Hexim1 would correspond to a self-hugging topology between the two monomers (Fig. 3F).

## PYNT motif is sequestered by autoinhibition and is unmasked upon RNA binding

The PYNT motif (conserved residues $^{202}$PYNTTQFLM$^{210}$) plays a critical role in directly binding and inactivating the Cdk9 kinase subunit of P-TEFb[7,21]. We further investigated the PYNT interactions in BR-L-AR using an A201C$^{MTSL}$ substitution which positions the MTSL-tag immediately N-terminal to the P$_{202}$YNT motif (Fig. 4A). The PRE profile of $^{15}$N BR-L-AR(A201C$^{MTSL}$) showed significant broadening of the PYNT motif itself, BR and c-AR (Fig. 4A and Supplementary Fig. 12). This result recapitulated the findings from BR-L-AR(S226C$^{MTSL}$) (Fig. 3B orange) and confirms that the PYNT motif is a central node in the BR–PYNT–AR inter-monomer network. Based on data suggesting that αL is involved (Fig. 3C), we next probed from its central region using a S183C$^{MTSL}$ tag (Supplementary Fig. 13). The PRE profile of $^{15}$N BR-L-AR(S183C$^{MTSL}$) showed significant broadening extending from the αL through the

PYNT motif excluding the C-terminal end of αL, which along with BR and the two ends of AR shows secondary broadening (Fig. 4B). This indicates that the inter-monomer BR–PYNT–AR interactions partially involve the central region of αL.

Lastly, we investigated the effect of 7SK RNA binding on the inter-monomer interactions. Addition of SL1-dI to $^{15}$N BR-L-AR(S183C$^{MTSL}$) resulted in significant peak intensity recovery across the entire protein, except for BR (Fig. 4B, cyan bars). This recovery signifies that the PRE-induced broadening has disappeared, meaning the inter-monomer network has been disrupted and the S183C$^{MTSL}$ tag is no longer in proximity to the rest of the protein. Note that the lack of intensity recovery for the BR is expected. As shown previously (Supplementary Fig. 9), BR residues are independently and severely broadened by the direct binding of SL1-dI RNA, masking any PRE-related effects. Taken together, the PRE data show that the PYNT motif directly participates in inter-monomer interactions, where it simultaneously contacts both BR and AR. This autoinhibitory sequestration is released upon BR binding to RNA, resulting in the unmasking of PYNT, making it available to bind and inactivate Cdk9 in the 7SK RNP.

## Linear 7SK SL1 contains two high-affinity BR binding sites

While binding of Hexim1 BR to SL1 Site1 has been well documented, other potential binding sites have been proposed but not characterized[24–26,30,43,44]. To resolve the outstanding questions of Hexim1–7SK RNA binding site location(s), stoichiometry, and specificity, we systematically investigated the full 7SK SL1 domain in linear versus circular conformations. Linear 7SK SL1 folds into a minimum 36-bp long stem containing eight internal bulges or loops (Fig. 5A). We

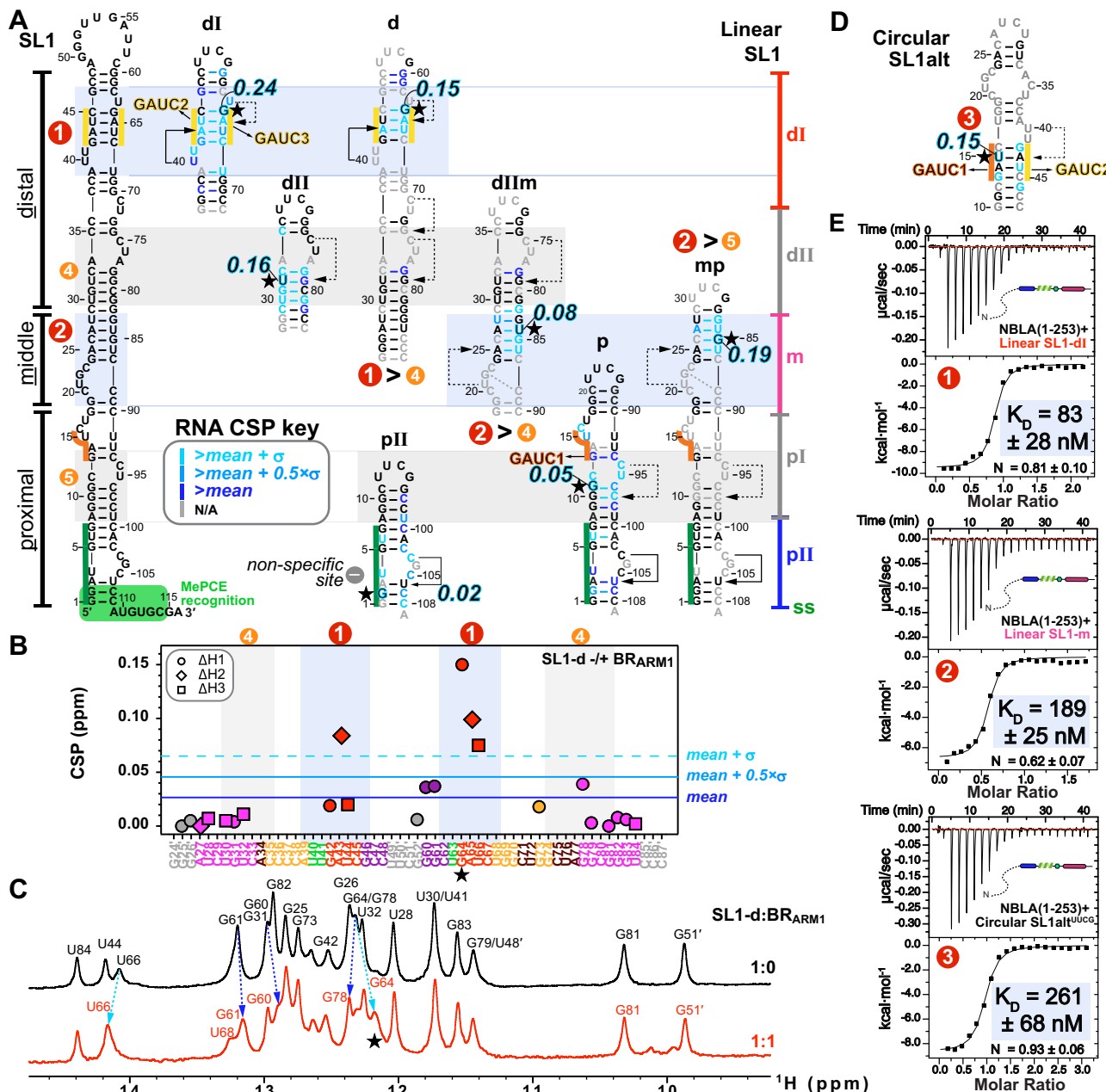

**Fig. 5 | Mapping and characterizing the 7SK RNA binding sites for Hexim1 Basic Region. A** Sequences and NMR-verified secondary structures of 7SK SL1 constructs aligned with the full-length SL1 secondary structure. Nucleotides with significant Chemical Shift Perturbation (CSP) upon binding Hexim1 BR_ARM1 are colored as indicated by the CSP key insert. The nucleotide experiencing the largest shift is highlighted with black/cyan font and a star, with their corresponding CSP values labeled. Severely broadened and highly overlapping resonances were excluded from the free/bound pair of CSP calculations. SL1-dI and SL1-dII CSPs shown were from BR-L-AR and BR-L titrations, which had a more complete set of RNA proton resonance assignments; all others are with BR_ARM1. Determined binding sites for Hexim1 monomeric constructs are numbered according to relative order of affinities and highlighted by light blue boxes (Sites 1, 2, 3) and gray boxes (Sites 4, 5),

respectively. Regions of SL1 are indicated as distal (dI and dII), middle (m), and proximal (pI and pII) as indicated on the right of panel A. GAUC motifs are highlighted with orange (GAUC1) and yellow (GAUC2/3) lines, and base-triples are indicated by connecting solid and dashed lines[25,26,44]. **B** CSP plot of SL1-d upon binding Hexim1 BR_ARM1. **C** 1D imino spectra of SL1-d in the absence and presence of BR_ARM1. **D** Sequence and NMR-verified secondary structure of circular SL1alt, with RNA CSP upon binding BR-L-AR indicated. **E** Isothermal Titration Calorimetry (ITC) measurements showing affinities of individual 7SK RNA Sites 1, 2 and 3 binding to Hexim1 NBLA. In panel (**E**), binding sites are numbered and highlighted as in panels A and D. A minimum of 3 independent ITC titration experiments were acquired for each sample.

employed a "divide-and-conquer" strategy, designing a series of RNA stem-loop segments from the distal (d), middle (m) and proximal (p) regions of linear SL1, as well as circular SL1alt. All constructs include a minimum 4-bp stem with one flanking bulge, based on the Hexim1 binding requirements for Site1 described above. We then used NMR chemical shift perturbations (CSPs) to map binding sites (Fig. 5) and

EMSA to determine stoichiometry (Supplementary Figs. 14, 15). For NMR mapping, BR_ARM1 was used in order to reduce the line broadening in the longer RNA constructs (SL1-d, dIIm, p and mp) when bound to protein. We verified that the same nucleotides in Site1 were perturbed for BR_ARM1 as for BR-L-AR (Fig. 5 and Supplementary Fig. 17D). NMR assignments, CSP plots and 1D spectra of titration series for the

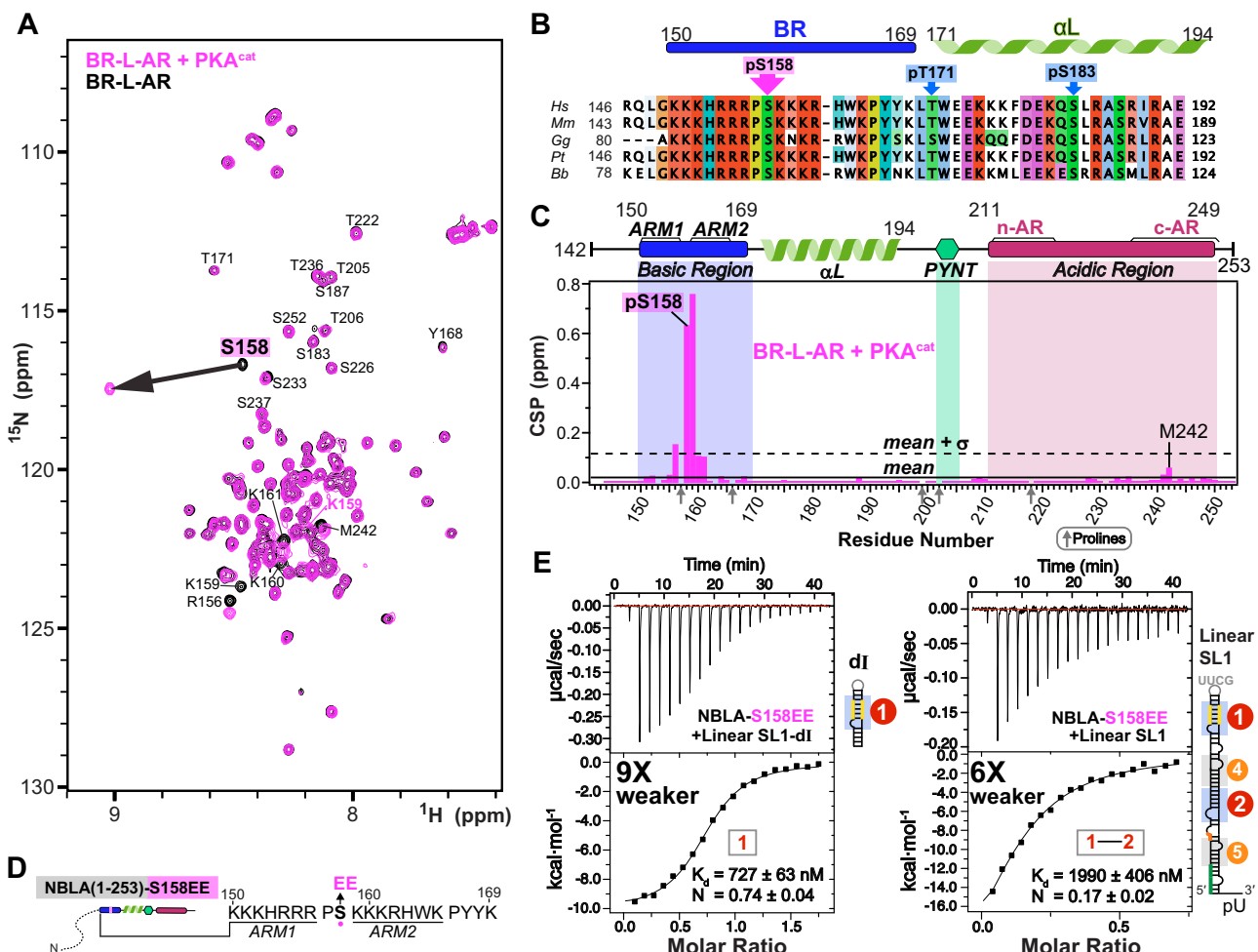

**Fig. 6 | S158 phosphorylation only minimally affects BR-L-AR autoinhibition, while S158EE impairs interaction with 7SK RNA. A** 600 MHz $^{15}$N-$^{1}$H HSQC spectral overlay of BR-L-AR without (black) and with protein kinase A catalytic subunit (PKA$^{cat}$) incubation at 37 °C for 3 hr (magenta). **B** Hexim1 BR and αL region sequence showing the location the three observed PKA$^{cat}$ phosphorylation sites. **C** Chemical Shift Perturbation (CSP) of Hexim1 BR-L-AR showing changes around pS158 and small but above average perturbations of c-AR residues F241 and M242. **D** Schematic of NBLA-S158EE phosphomimetic substitution. **E** Representative Isothermal Titration Calorimetry (ITC) measurements of NBLA-S158EE binding to SL1-dI (left) and full-length SL1 (right). ITC measurements for each sample were acquired with a minimum of 3 independent titration experiments. The affinity fold changes compared to wild-type NBLA are indicated next to each ITC fitting plot.

additional RNA constructs studied (Supplementary Figs. 16–22) were acquired as for Site1. Using the RNA CSPs (Fig. 5A–D), we mapped binding Sites 1, 2, 4 and 5 in linear SL1 and Site3 in circular SL1alt. The non-specific site at SL1-pII exhibits minimum CSP upon binding BR$_{ARM1}$, providing a negative control. The binding sites are numbered based on the relative order of binding affinities observed by NMR, where Site1 (not Site4) shifted by adding 1:1 BR$_{ARM1}$ to SL1-d, Site2 (not Site4) shifted in SL1-dIIm, and Site2 (not Site5) shifted in SL1-mp (Fig. 5A); and by EMSA experiments, where two sequential shifts were observed for SL1-d$_{\Delta U}$ and SL1-d, corresponding to the high-affinity Site1 followed by low-affinity Site4 (Supplementary Fig. 14D, E and Notes). The largest CSP values and the exchange behaviors on the NMR timescale also agreed well with the order of binding affinities determined from combined NMR and EMSA analysis (Fig. 5A stars and Supplementary Fig. 17C).

To quantitatively compare the binding affinities of these RNAs, we acquired thermodynamic binding parameters by ITC (Figs. 5E, 6). Due to sample precipitation that occurs during ITC titration of RNA into BR-L-AR, potentially due to stirring, we instead used N-BR-L-AR (NBLA hereafter) that includes the N-terminal region. ITC measurements yielded K$_D$ values of 83, 189, and 261 nM for Sites 1, 2 and 3, respectively. Together, these findings indicate that linear 7SK SL1

contains two high-affinity sites and two low-affinity sites for BR. In contrast, circular 7SK SL1alt contains only one binding site for one BR, at the alternative GAUC1-GAUC2 palindrome (Fig. 5D). Comparison of these 5 binding sites indicates that the determining factor for high-affinity Hexim1 binding appears to be an A-U rich stem with at least one flanking 5′ U-containing bulge that participates in a U⋯A-U base triple, a common feature for Sites 1 and 2, but not for Sites 4 and 5, or the non-specific site SL1-pII. Site3 can form a U$_{40}$⋯A$_{43}$-U$_{15}$ base triple, but the U$_{40}$U$_{41}$ bulge is now on the 3′ side of the GAUC1-GAUC2 stem instead of on the 5′ side in Site1, geometrically analogous to the U$_{63}$ bulge of Site1, consistent with a lower affinity than Sites 1 and 2.

## S158 phosphorylation minimally affects autoinhibition and S158EE impairs RNA binding

Multiple phosphorylation sites have been identified in Hexim1 unstructured regions and suggested to play roles in releasing P-TEFb from the 7SK RNP complex[51–54], including phosphorylation of BR residue Ser158 (pSer158) by protein kinase C (PKC)[55] or protein kinase A (PKA)[56]. To investigate the effect of pSer158 on Hexim1 conformation, we phosphorylated BR-L-AR in vitro using PKA catalytic subunit (PKA$^{cat}$) and monitored phosphorylation by NMR $^{1}$H-$^{15}$N HSQC (Fig. 6A,

**A**

| | Sites | K_D (nM) | N |
|---|---|---|---|
| **NBLA monomer** | | | |
| SL1-dI | 1 | 83 ± 28 | 0.81 ± 0.10 |
| SL1-m | 2 | 189 ± 25 | 0.62 ± 0.07 |
| SL1 | 1,2 (4,5) | 336 ± 134 | 0.36 ± 0.01 |
| SL1alt | 3 | 261 ± 68 | 0.93 ± 0.06 |
| *S158EE* | | | |
| SL1-dI | 1 | 727 ± 63 | 0.74 ± 0.04 |
| SL1 | 1,2 (4,5) | 1990 ± 406 | 0.17 ± 0.02 |
| **Hexim1 dimer** | | | |
| SL1-dI | 1 | 780 ± 25 | 0.76 ± 0.01 |
| SL1-m | 2 | 885 ± 58 | 0.79 ± 0.07 |
| SL1 | 1,2 (4,5) | 73 ± 30 | 0.39 ± 0.05 |
| SL1alt | 3 | 2040 ± 359 | 0.68 ± 0.04 |
| SL1-dII | 4 | 1220 ± 129 | 0.47 ± 0.05 |
| SL1-p | 5 | 2210 ± 197 | 0.49 ± 0.03 |
| *A3m* | | | |
| SL1-p | 5 | 276 ± 17 | 0.34 ± 0.03 |
| *A12m* | | | |
| SL1-p | 5 | 396 ± 111 | 0.10 ± 0.01 |
| *ΔPYNT* | | | |
| SL1-p | 5 | 451 ± 262 | 0.11 ± 0.02 |
| **Hexim1 dimer (syringe)** | | | |
| SL1-dI | 1 | 458 ± 28 | 1.12 ± 0.05 |
| SL1-m | 2 | 782 ± 26 | 0.87 ± 0.04 |
| SL1 | 1,2 (4,5) | 183 ± 18 | 1.75 ± 0.01 |
| SL1alt | 3 | 748 ± 15 | 1.01 ± 0.09 |
| SL1-d(api) | 1,4 | 96 ± 22 | 1.40 ± 0.08 |

**B**

**C**

**Fig. 7 | ITC studies for NBLA monomer versus Hexim1 dimer with 7SK RNA constructs reveal that autoinhibition renders specificity for dual high affinity sites in linear 7SK. A** Table summarizing thermodynamic parameters for the binding of Hexim1 to 7SK RNA determined by Isothermal Titration Calorimetry (ITC). Protein variants of NBLA monomer and Hexim1 dimer are highlighted in italicized fonts. All values are an average of a minimum of 3 independent titration experiments, and standard deviations are reported as ± value. The order of titration is RNA in the syringe to protein in the sample cell, with the exception of the bottom section for Hexim1 dimer (syringe), where the titration order is reversed. Circular SL1alt in the ITC experiments refers to the UUCG tetraloop construct. **B** Cartoon illustrations of NBLA monomer binding to full-length linear SL1 versus RNA constructs with a single site. **C** Cartoon illustrations of Hexim1 dimer binding to full-length linear SL1 versus RNA constructs with a single site. For simplicity, the αL helix is not shown as curved.

B). A large shift of S158 amide resonance was observed due to pS158, with induced shifts at adjacent lysines 159–161 and R156 (P157 lacks an amide) (Fig. 6A, C). Of note, c-AR residues F241 and M242 also show small chemical shift changes, although to a much smaller extent (CSP ~10-fold smaller than in Fig. 2C). This suggests that pS158 only minimally induces the same allosteric change in c-AR as RNA binding does, largely maintaining autoinhibition.

Longer incubation with PKA resulted in phosphorylation of additional sites (T171 and S183) and destabilization of the entire αL (Supplementary Fig. 23A). We note that αL does contribute to RNA binding affinity, as shown by our EMSA (compare Supplementary Fig. 14C to 14A). To avoid potential destabilization of αL and assure a homogenous sample, as well as to avoid a predicted pS49 by PKA (Supplementary Fig. 23B), we made a phosphomimetic variant NBLA-S158EE to look at the effect of two additional negative charges in BR on RNA binding by ITC, in the context of the longer soluble monomeric construct. An increase in K_D of 9- and 6-fold was observed for NBLA-S158EE with SL1-dI and full-length linear SL1, respectively, compared to wild-type NBLA (Fig. 6D, E), indicating that S158EE significantly reduces RNA binding. Together, these observations suggest that in vivo S158 phosphorylation would reduce RNA binding without significantly disrupting Hexim autoinhibition, thereby triggering release of P-TEFb as a net result.

## Both monomers in Hexim1 dimer engage in high-affinity RNA binding simultaneously

Since full-length Hexim1 is a stable homodimer, the above results with Hexim1 monomeric constructs suggest that a dimer could simultaneously bind two sites on linear SL1, one for each monomer. Indeed, EMSAs using full-length SL1 or the minimum functional 7SK (SL1-SL4, that has been shown to assemble in cells with MePCE−Larp7−Hexim−P-TEFb[57]) show that full-length Hexim1 dimer binds first to Site1 and Site2. At higher Hexim1:RNA ratios, another dimer subsequently binds to Sites4 and Site5 (Supplementary Fig. 14F,G). These Site4 and Site5 are likely not primary binding sites for Hexim1 in cells, rather they may serve as docking sites during the Hexim1 searching process for Sites 1 and 2.

We next used the same short SL1 constructs described above versus full-length SL1, titrated into Hexim1 NBLA monomer versus full-length Hexim1 dimer, to determine binding stoichiometry by ITC (Fig. 7 and Supplementary Fig. 24). Systematic comparison of the N values shows that the monomeric NBLA binds as one molecule per RNA to individual Sites 1, 2, and 3 ($N = 0.62$–$0.93$) and binds as two molecules per full-length linear SL1 ($N = 0.36$) (Fig. 7A, B). Full-length Hexim1 (dimer) also binds as one monomer per individual Sites 1, 2, and 3 ($N = 0.68$–$0.79$, using monomer concentrations when fitting titration curves), and binds as two monomers, i.e., one dimer, per

linear SL1 ($N = 0.39$) (Fig. 7A, C). We further verified the stoichiometry by reversing the titration order (i.e., titrated Hexim1 from the syringe to RNA in the cell), and again observed one monomer per single-site RNAs ($N = 0.87$–$1.12$) and two monomers (one dimer) for linear SL1 ($N = 1.75$). Together with the NMR and EMSA results, we conclude that both monomers within Hexim1 dimer are engaged in SL1 RNA binding, to Sites 1 and 2, simultaneously (Fig. 7C).

### Hexim1 autoinhibition renders specificity for linear 7SK SL1

After mapping the individual RNA binding sites with their relative order of affinities and determining the stoichiometry of Hexim1 dimer binding with SL1, we next compared the ITC-determined binding affinities between NBLA and Hexim1 dimer, where inter-monomer autoinhibition is enforced by dimerization in the full-length Hexim1 but not in the monomer, to quantitatively determine the effect of Hexim1 autoinhibition on RNA binding. Comparison of the dissociation constants $K_D$ shows that the Hexim1 dimer binds 10-, 5- and 8-fold weaker than the monomer (NBLA) to the individual Site1, Site2, and Site3, respectively, suggesting that autoinhibition of Hexim1 dimer weakens binding to single sites (Fig. 7). However, Hexim1 dimer binding to linear SL1 shows cooperative binding with a > 10-fold lower $K_D$ (73 nM) than the single Site1 and Site2 (780 and 885 nM). In contrast, no cooperativity was observed for NBLA monomer, with a $K_D$ value for SL1 (336 nM) slightly larger than those of Site1 and Site2 (83 and 189 nM). This cooperative effect that is exclusively observed for the dimer is recapitulated in experiments where the titration order is reversed, albeit with a smaller effect likely due to increased aggregation when Hexim1 is concentrated in the syringe (Fig. 7A bottom). These results indicate that the autoinhibition of Hexim1 dimer is required for the specificity towards full-length linear SL1 over single RNA sites, whereas NBLA monomer lacks this specificity.

The weaker single sites SL1-dII (Site4) and SL1-p (Site5) exhibited larger $K_D$ values (1221 and 2208 nM, respectively) than Site1 or Site2 for Hexim1 dimer. Interestingly, the N values for these weaker binding sites are slightly smaller (0.47 and 0.49), indicating a partial release of autoinhibition such that only one monomer is available for binding within a subpopulation of the dimer. In contrast, SL1alt (Site3) in circular 7SK conformation has weak binding affinity to Hexim1 dimer (2035 nM), albeit with a similar N value (0.68) to Sites 1 and 2 (0.76 and 0.79). This Hexim1 dimer–SL1alt affinity is unexpectedly much lower than the NBLA monomer–SL1alt, suggesting significantly different binding kinetics for dimer–Site3 from monomer–Site3. Consistent with these results, a SL1-d(api)[27] (identical to SL1-d except for the apical loop sequence) has an enhanced binding affinity (96 nM) but a smaller N value (1.40) than SL1, indicating two binding events overall (Sites 1,4 vs 1,2) but with a moderately reduced release of Hexim1 autoinhibition.

Finally, to further test the role of Hexim1 autoinhibition in RNA-binding specificity, we disrupted the autoinhibition with three Hexim1 dimer variants, A3m and A12m, that neutralized negatively charged residues in c-AR and n-AR with alanines[7], and ΔPYNT, and titrated them with the low-affinity binder SL1-p (Site5) monitored by ITC (Fig. 7 and Supplementary Fig. 25). These three variants showed 8-, 6- and 5-fold tighter binding affinities than wild-type Hexim1, respectively, suggesting that disruption of autoinhibition turned Hexim1 into a non-specific RNA binder.

Together, these results show that Hexim1 dimer preferentially and cooperatively binds Sites 1 and 2 in linear SL1 with each monomer, respectively, and the inter-monomer autoinhibition within the dimer renders the specificity for linear SL1 over isolated RNA binding sites, including the circular SL1alt. This is achieved through two sides of the same coin: cooperative binding to the dual high-affinity sites in SL1 and decreased affinity for RNAs with only a single binding site. Together with the PRE results, we conclude that Hexim1 binding to two sites in linear SL1 provides an effective allosteric switch to fully release Hexim1

autoinhibition in the dimer, which subsequently exposes two PYNT motifs for Cdk9 binding and inactivation from two P-TEFb heterodimers (Fig. 8).

## Discussion

RNA-binding intrinsically disordered protein regions (IDRs) are increasingly recognized as important and versatile regulators of RNA metabolism, RNP phase separation and cellular signaling, yet their mechanistic roles remain elusive compared to their well-structured counterparts[58,59]. Recent advances have also uncovered that such IDRs can modulate the RNA- and DNA-binding of an adjacent folded domain[60,61]. Our studies shed light on this growing paradigm by dissecting how the Hexim1 disordered BR-L-AR enables finely tuned interaction with the conformationally dynamic non-coding RNA, 7SK.

We show that Hexim1 employs a stand-alone disordered basic region (BR) to selectively recognize A-U-rich RNA stems with flanking U-rich bulge(s), whereas the disordered acidic region (AR) serves as a specificity checkpoint. In the absence of RNA, the Hexim1 central region adopts a conformational ensemble wherein the key regulatory PYNT motif, required for Cdk9 binding and inactivation, becomes masked through a BR–PYNT–AR interaction network—specifically in a head-to-tail inter-monomer configuration (Fig. 8). Strikingly, 7SK RNA binding to the BR disrupts this autoinhibition, effectively unmasking PYNT and enabling subsequent interaction with Cdk9. Using a monomeric BR-L-AR construct, we demonstrate that linear 7SK SL1 harbors two high-affinity and two low-affinity binding sites (Sites 1,2 versus 4,5). Notably, Hexim1 dimer binds linear SL1 with a 1-to-1 stoichiometry, with each monomer BR engaged with a high-affinity site. These findings, together with previous studies[9,22], support a model in which Hexim1 CC dimerization enables BR-L-AR mediated inter-monomer dual autoinhibition (Fig. 8): BR sequestration restricts RNA binding, while PYNT masking prevents premature Cdk9 interaction, and both are reversed upon specific RNA recognition of two high-affinity sites.

The regulatory role of AR is thus two-fold towards RNA binding: (1) autoinhibition enhances specificity towards linear 7SK SL1 over its circular SL1alt; and (2) autoinhibition prevents Hexim1 from being trapped by non-specific binding to RNAs, accelerating the search for functional 7SK targets—an RNA-analog to the "facilitated search" role recently described in disordered D/E patches for DNA-binding protein[62]. In light of this particular feature of Hexim1, our study illustrates the importance of considering Hexim1 dimer interaction with full-length linear SL1 and circular SL1alt, in contrast to previous foundational studies that primarily focused on a single site (Site1)[25–27,44]. Our results provide a more complete picture for 7SK RNA engagement, highlighting how 7SK conformational dynamics influence the regulatory architecture of 7SK RNP. We show that Hexim1 dimer binds circular 7SK SL1alt, which has a single binding site, with micromolar affinity, but binds linear 7SK SL1 with nanomolar affinity (73 nM; Fig. 7C), reinforcing RNA conformational switching as a central P-TEFb release mechanism[35].

Building on our comprehensive binding studies and prior structural studies of MePCE–7SK–Larp7 core RNP[31,34,63], we propose a refined stoichiometry model for the P-TEFb-inhibitory 7SK RNP: one linear 7SK RNA, one MePCE, one Larp7, one Hexim dimer bound to two RNA sites, and two P-TEFb heterodimers[64] (Fig. 8). The ability of Hexim1 to simultaneously inactivate two P-TEFb complexes suggests a signal-amplifying module, where two P-TEFb units can be activated per unit cellular signaling event. This model also implies a cooperative assembly mechanism, in which 7SK RNA, Hexim and P-TEFb mutually stabilize each other in 7SK RNP, contributing to a fast and efficient on- and off-switch during the Pol II pause release and the proposed transcription termination via 7SK[65], respectively.

For protein-centric P-TEFb release pathways, we show here that S158 phosphorylation acts as an RNA-binding off-switch, while

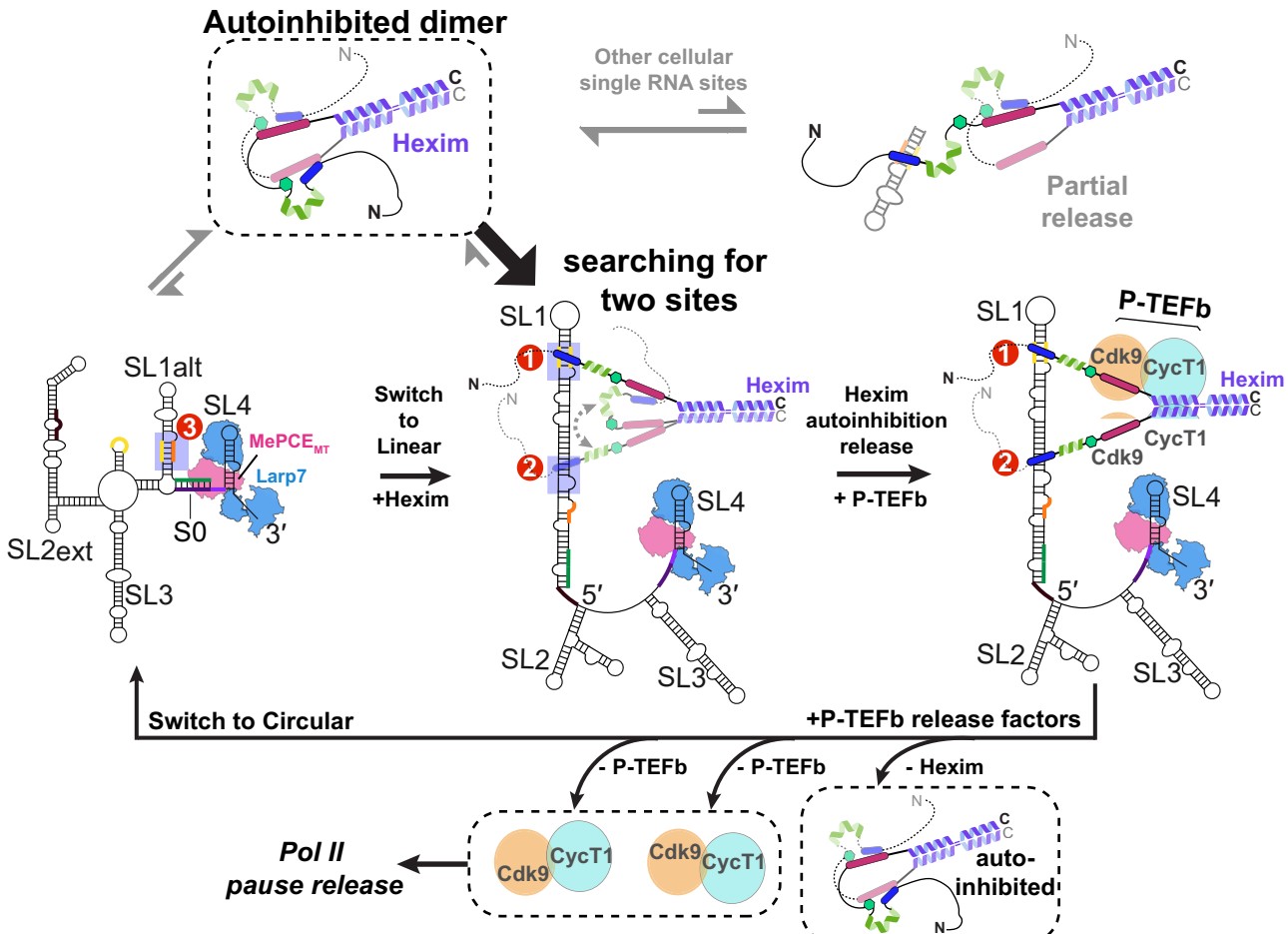

**Fig. 8 | Proposed model of how Hexim1 autoinhibition governs the specificity for linear 7SK SL1 binding in order to expose PYNT motifs.** In the full-length Hexim1 homodimer, inter-monomer interactions mask the PYNT motif using both BR and AR (top left). Non-functional RNA sites (top right) or circular 7SK (bottom left) do not stably release the autoinhibited Hexim1 dimer. A complete set of two high-affinity RNA-binding sites in linear 7SK releases the Hexim1 autoinhibition and exposes PYNT motifs for P-TEFb inactivation in the assembled 7SK RNP. A switch back to circular 7SK by P-TEFb release factors will coincide with Hexim1 dimer going back to its autoinhibited state, and consequent release of two units of P-TEFb for subsequent stimulation of Pol II transcription pause release.

maintaining Hexim1 autoinhibition, thereby releasing P-TEFb. These results highlight the need for further investigations of Hexim post-translational modifications in both RNA binding and autoinhibition. Comparison of Hexim1 with its paralog Hexim2 shows that Hexim2 has a much shorter AR, which lacks the c-AR found in Hexim1 (Supplementary Fig. 2). Our data suggest that c-AR plays a critical role in the BR–PYNT–AR interaction network. Thus, Hexim2 may exhibit reduced autoinhibition and altered P-TEFb regulatory dynamics, particularly in tissues where Hexim2 is preferentially expressed.

In addition to linear-to-circular 7SK conformational switching, another RNA-centric pathway for regulated P-TEFb release was recently shown, i.e., N6-methylationadenosine (m6A) modifications of 7SK RNA by METTL3-METTL14 alters Hexim binding. Specifically, m6A43 and m6A65 in GAUC2 and GAUC3 motifs, which are more prominent in cancer cells[66,67], likely disrupt Site1 base-triples essential for Hexim binding[44]. This would diminish Site1 affinity and effectively convert linear SL1 into a one-site binder (Site2), reducing Hexim1 dimer binding and promoting P-TEFb release. A similar strategy is adopted by HIV-1[16,68], where Tat protein only binds Site1 but not Site2[30], and lacks an analogous acidic patch that could inhibit its single ARM (Supplementary Fig. 26). However, details of the molecular mechanism of Hexim–Tat competition to form the intermediate Tat–7SK RNP at HIV-1 promoter remain elusive[14,69]. Notably, past studies of Tat–Hexim competition focused only on Site1[26,27,43]; our results suggest that inclusion of Site2, along with consideration of the autoinhibitory state

of Hexim dimer, is crucial for understanding the molecular basis of this competition.

In summary, this work resolves three major longstanding questions about Hexim function in 7SK RNP: how autoinhibition enforces Hexim1 specificity towards linear 7SK RNA; the precise stoichiometry and architecture of Hexim–7SK binding and the minimal components of the P-TEFb-inhibitory 7SK RNP; and the mechanism by which Hexim1–7SK binding unmasks the PYNT motif, enabling Cdk9 sequestration and inactivation in the 7SK RNP. These insights provide a mechanistic foundation for understanding the regulation of P-TEFb binding and release by 7SK RNP, which is fundamental to its role in eukaryotic transcription. They also open previously unexplored avenues for exploring how various P-TEFb release factors and HIV-1 Tat could interact with and/or remodel 7SK RNA conformational dynamics.

## Methods

### Plasmids and mutagenesis

Full-length human Hexim1 protein (aa 1–359) and monomeric Hexim1 BR-L-AR (aa 142–253) were cloned into pETDuet-1 vector with an N-terminal His6-tagged maltose binding protein (MBP) fusion, followed by Tobacco Etch Virus (TEV) protease cleavage site between MBP and Hexim1. Further truncated monomeric Hexim1 constructs, BR-L (aa 136–209) and BRext (aa 149–179) were cloned into pETDuet-1 vector with N-terminal His6-tagged 56-residue B1 domain of

Streptococcal protein G (GB1) fusion[70] followed by Tobacco Etch Virus (TEV) protease cleavage site between GB1 and Hexim1 monomers. BR (aa 141–165) and BR$_{ARM1}$ (aa 141–159) constructs were cloned by site-directed mutagenesis using the pETDuet-1 GB1-ts-BR-L plasmid as template. A non-native glycine remains at the N-terminus of the final full-length Hexim1 protein, BR-L-AR, BR-L, BR and BR$_{ARM1}$ after TEV protease cleavage, whereas a native Glycine residue from BRext (aa 149) was a part of the TEV cleavage site (ts). Cysteine substitutions of BR-L-AR variants for PRE (S183C, A201C, S226C, S233C and S237C) were cloned by site-directed mutagenesis using the pETDuet1 MBP-ts-BR-L-AR as template.

Minimum linear 7SK (SL1-SL4$^{ms2}$) and linear SL1 RNA genes were cloned into the pUC19 vector with a Hammerhead ribozyme gene at the 3′ end followed by a BamHI restriction enzyme cleavage sequence. All site-directed mutagenesis was performed using the Q5 kit (New England Biolabs).

## Protein expression and purification

Hexim1 protein genes in either pETDuet-MBP or pETDuet-GB1 vectors were transformed into BL21-GOLD(DE3) competent cells for protein expression. Bacterial culture was grown at 37 °C in minimum media to O.D.600 of 0.6, then transferred to 18 °C for 1 hr prior to induction by 0.5 mM IPTG for 18–20 hrs. For uniformly [$^{15}$N, $^{13}$C] (or [$^{15}$N]) enriched Hexim1 proteins, M9 minimal media containing 1 g/L of $^{15}$NH$_4$Cl and 3 g/L of [$^{13}$C-6]-D-glucose (or natural abundance glucose) (Cambridge Isotope Laboratories) were used. Cells were pelleted by centrifugation, resuspended in lysis buffer (20 mM Tris, pH 7.5, 1 M NaCl, 5 mM imidazole and 1 mM PMSF) supplemented with protease inhibitor cocktail tablet (ThermoScientific Pierce A32965) and lysozyme. Cells were lysed by sonication on ice and clarified by centrifugation and filtration with 0.45 μm syringe filter. Lysates were loaded onto a 5-mL Ni Sepharose affinity column (HisTrap-HP; GE Healthcare), and a linear gradient of 5–500 mM imidazole was used to elute the His$_6$-tagged proteins. The eluted protein was added 1 mg of TEV protease, dialyzed (20 mM Tris, pH 7.5, 300 mM NaCl, 5 mM βME) and further purified with a second Ni affinity column to removed His$_6$-tagged fusion proteins (MBP or GB1) and TEV protease. For the full-length Hexim1 protein, an additional purification step was carried out with size-exclusion chromatography (SEC; HiLoad 26/600 Superdex 200 or HiLoad 16/600 Superdex 75; GE Healthcare) in binding buffer (20 mM HEPES, pH 7.5, 150 mM KCl, 1 mM TCEP). Proteins were concentrated using Amicon devices (Millipore), and concentration was measured by absorbance at 280 nm. For monomeric Hexim1 BR-L-AR, concentrating and buffer exchanging using Amicon devices into a final buffer (50 mM sodium phosphate, pH 6.2, 0/150/300 mM KCl, 1 mM TCEP, 0.02% NaN$_3$, 5% D$_2$O added post buffer exchange for NMR) was performed with caution to prevent precipitation at higher local protein concentration. For even shorter monomeric proteins, 1 kDa MWCO dialysis tubing (for BRext) or desalting column (HiTrap Desalting 5 mL, GE Healthcare; for BR and BR$_{ARM1}$) were used for buffer exchange into 5 mM NH$_4$HCO$_3$ buffer, freeze dried with a lyophilizer and dissolved directly in NMR buffer (50 mM sodium phosphate, pH 6.2, 150 mM KCl, 0.02% NaN$_3$, 5% D$_2$O).

## In vitro phosphorylation by PKA catalytic subunit

Phosphorylation of Hexim1 BR-L-AR was performed utilizing cAMP-dependent protein kinase A catalytic subunit (PKA$^{cat}$; NEB P6000S). The protein was first buffer-exchanged into NMR buffer (50 mM sodium phosphate, pH 6.2, 150 mM KCl, 0.02% NaN$_3$, 5% D$_2$O), then concentrated with constant mixing using Amicon devices (Millipore). The final phosphorylation reaction (500 μL total volume) contained 35 μM BR-L-AR, 2 mM ATP, 10 mM MgCl$_2$, I mM EDTA, 2 mM DTT, and 4 μL PKA (10,000 units). Two separate reactions were incubated in a water bath at 37 °C for 3 and 24 h, respectively, before being analyzed using NMR.

## NativePAGE and mass photometry characterization of Hexim1 constructs

Prior to loading, freshly purified protein samples of full-length Hexim1, L-AR-CC, and BR-L-AR were centrifuged at 14,000 × $g$ for 10 minutes at 4 °C to remove potential contaminants and aggregates. Samples were prepared with NativePAGE™ Sample Buffer (4X) (Invitrogen BN2003). For full-length Hexim1 and L-AR-CC, 5 μg was loaded per lane. For BR-L-AR, 2.5 μg and 5 μg were loaded on separate lanes. Samples were run on NativePAGE™ 3–12% Bis-Tris Mini Gels (Invitrogen™ BN1003BOX) at room temperature at 150 V using a cathode buffer supplemented with Coomassie Blue G-250 at a final concentration of 0.02% (Invitrogen™ BN2007).

Mass photometry measurements of full-length Hexim1, L-AR-CC, and BR-L-AR were performed on a Refeyn TwoMP mass photometer. Freshly purified protein samples were prepared as 4 μM for monomers and 2 μM for dimers, diluted (or mixed as complexes and diluted) with SEC buffer to 400 nM stocks, and then added by the drop-dilution method into phosphate-buffered saline (137 mM NaCl, 2.7 mM KCl, 10 mM Na$_2$HPO$_4$, 1.8 mM KH$_2$PO$_4$, pH 7.4) to achieve final total protein concentrations of 40 nM. Interferograms were recorded for 60 seconds with Refeyn AcquireMP, and data were analyzed with Refeyn DiscoverMP using three molecular weight standards for calibration: Bovine Serum Albumin (BSA, ThermoFisher 23209) 66 kDa/132 kDa, beta-amylase (BAM, Sigma A8781) 112 kDa/224 kDa and Bovine Thyroglobulin (TG, Sigma 609310) 670 kDa (R$^2$ = 1.000, Max. mass error 3.0%). The percentages associated with each peak fitting are derived from the DiscoverMP analysis, which calculates the proportion of total particle counts represented by the Gaussian-fitted peaks.

## Nitroxide spin labeling of proteins

Protein modification with nitroxide spin label was carried out using the established protocol[71]. Briefly, each purified protein solution, containing about 50 nmole of Hexim1 BR-L-AR cysteine variants, was buffer exchanged using a PD-10 desalting column with the gravity protocol to remove the reducing agent into PRE buffer (50 mM sodium phosphate, pH 6.2, 150 mM KCl, 0.02% NaN$_3$). The PD-10 elution was collected into a falcon tube, containing 10-fold molar excess of S-(1-oxyl-2,2,5,5,-tetramethyl-2,5,-dihydro-1H-pyrrol-3-yl) methylmethanesulfonothiolate (MTSL; Toronto Research Chemicals TRC-O875000) dissolved in 0.5 mL of PRE-buffer, covered with aluminum foil to avoid light. The reaction was kept at room temperature under gentle stirring in the dark for > 15 min. Subsequently, an additional aliquot of MTSL was added to the reaction mixture to reach a final ratio of 1:20 protein:MTSL. The reaction was further incubated at room temperature under gentle stirring overnight in the dark.

The unreacted MTSL molecule was removed by running through a PD-10 desalting column equilibrated in PRE buffer twice using the gravity protocol. The MTSL-tagged protein was concentrated with caution using Amicon devices, and 5% v/v D$_2$O was added to prepare the final NMR sample.

## In vitro transcription of RNA

RNA samples were prepared by in vitro transcription with T7 RNA polymerase P266L variant[34,72] and are described briefly below. Short RNA constructs, including linear 7SK SL1 dissected constructs SL1-dI, SL1-dI$_{ΔU}$, SL1-dII, SL1-d, SL1-d$_{ΔU}$ SL1-dIIm, SL1-m, SL1-mp, SL1-pII, SL1-pIIo, SL1-p, and circular 7SK SL1alt (wt or UUCG apical loop), were transcribed from chemically synthesized DNA oligonucleotides (Integrated DNA Technologies). Transcription mixtures (40 mM Tris, pH 8.0, 1 mM spermidine, 0.01% Triton-X100, 2.5 mM DTT, 25 mM MgCl$_2$, 0.5 μM oligonucleotide DNA template, T7 RNAP P266L, 4 mM each rATP, rUTP, rGTP and rCTP) were incubated at 37 °C for 4–6 hrs. Longer RNA constructs, minimum linear 7SK (SL1-SL4$^{ms2}$) and linear 7SK SL1, were transcribed from plasmid DNA templates prepared with QIAGEN Plasmid Maxi Kit and linearized with BamHI-HF restriction

enzyme (New England Biolabs). Transcription mixtures are the same as the short RNA constructs above, except from lower MgCl$_2$ concentration (15-20 mM) and overnight incubation at 37 °C.

Transcription mixtures were loaded onto 10–20% denaturing polyacrylamide gel electrophoresis (Urea-PAGE; 19:1 crosslinking ratio) for RNA purification. Purified RNA molecules were eluted out of gel pieces either with an Elutrap device (GE Whatman) or by crush-and-soak in 1X TBE buffer (90 mM Tris-borate, 2 mM EDTA, pH 8.3). Collected RNA eluents were concentrated and buffer exchanged into sterilized nanopure water using 3 kDa or 10 kDa MWCO Amicon devices (Millipore), supplied with counterion in high salt buffer (10 mM sodium phosphate, pH 7.6, 1 mM EDTA, 1.5 M KCl) and buffer exchanged back into sterilized nanopure water. Dilute RNA samples < 100 µM were heated at 95 °C for 5 min and snap cooled on ice for 1 hr prior to final buffer exchanging and concentrating with Amicon devices (Millipore). Stock RNA solutions were exchanged into sterilized nanopure water and stored in − 20 °C freezer.

### Electrophoretic mobility shift assay

1 µL of 50 µM RNA and stock solutions of 25–50 µM Hexim1 proteins were used to achieve varying protein-to-RNA molar ratios in a total volume of 5.5 µL in binding buffer, and added 1 µL loading dye (30% glycerol and bromophenol blue). The binding mixtures were incubated at room temperature for 15 min prior to gel electrophoresis on 5.7% non-denaturing polyacrylamide gel (PAGE; 37.5:1 crosslinking ratio) or 0.8% Agarose gel (for minimum linear 7SK SL1-SL4$^{ms2}$) in 0.5X TBE buffer at room temperature. Polyacrylamide gels (BioRad Mini-PROTEAN handcast) and agarose gel (BioRad Mini-Sub 7 × 7 cm) were run at 70 V. Polyacrylamide gels were subsequently stained with Toluidine blue solution and destained in water, whereas agarose gels were made with SYBR™ Safe stain (Invitrogen) added prior to solidifying. Gels were imaged with a Pharos FX Plus scanner (Bio-Rad), and band intensities were extracted using ImageJ and fit to binding models (see Supplementary Fig. 14 notes) with Igor-Pro 9 software.

### NMR spectroscopy

For NMR assignments of SL1-dI$_{\Delta U}$ RNA bound BR-L-AR, dilute BR-L-AR uniformly enriched with $^{15}$N and $^{13}$C were combined with dI$_{\Delta U}$ in a 1:1.2 molar ratio, and the dilute complex was buffer exchanged in NMR buffer (50 mM sodium phosphate, pH 6.2, 150 mM KCl, 0.02% NaN$_3$, 5% D$_2$O) and concentrated down to a final concentration of 0.22 mM BR-L-AR and 0.25 mM SL1-dI$_{\Delta U}$. Backbone assignments were carried out using the following NMR experiments: HNCACB, CBCA(CO)NH, HNCA and C(CO)NH. For NMR assignments of Hexim1 BR-L-AR, 70 µM sample uniformly enriched with $^{15}$N and $^{13}$C in the NMR buffer was used. Backbone assignments were carried out using the CBCA(CO)NH and HNCA NMR experiments, which were collected with non-uniform sampling and processed with iterative soft thresholding reconstruction[73], and comparison to the RNA-bound BR-L-AR assignments.

For NMR assignments of further truncated Hexim1 constructs, BR-L, BR, BRext and BR$_{ARM1}$, samples uniformly enriched with $^{15}$N and $^{13}$C in the range of 0.5-0.8 mM concentration in NMR buffer were used. Backbone assignments were carried out using the following NMR experiments: HNCACB, CBCA(CO)NH and C(CO)NH. Additional side-chain assignments were acquired for BR-L and BR using HCCH-TOCSY. BR$_{ARM1}$ $^{15}$N-$^1$H HSQC resonances were assigned based on almost complete overlapping with the BR $^{15}$N-$^1$H HSQC peaks. For 1H imino sequential assignments of RNA samples in the absence and presence of monomeric Hexim1 proteins, the two-dimensional NOESY spectra were recorded using 5% D$_2$O samples at 283.15 K with NOE mixing time of 150 ms. For PRE NMR experiments, $^{15}$N-$^1$H HSQC spectra were recorded at 290.15 K with Avance III HD 600 MHz.

For NMR assignments of 7SK RNA constructs, natural abundance samples in the range of 0.15–1.24 mM concentration in NMR buffer

(50 mM sodium phosphate, pH 6.2, 150 mM KCl, 0.02% NaN$_3$, 5% D$_2$O, with the exception of 0 mM KCl NMR buffer for SL1-dI$_{\Delta U}$, SL1-dII and SL1-d, and 10 mM sodium phosphate, 50 mM KCl, pH6.3, 5% D$_2$O for SL1alt$^{UUCG}$) were used. For secondary structure determination and N-H...N imino proton assignments (exchangeable-proton, here H1 for Gua and H3 for Ura), NOESY and TOCSY spectra were recorded using 95% H$_2$O/5% D$_2$O samples at both 283 K and 298 K. For nonexchangeable proton assignments (here H2/H8 for Ade, H8 for Gua, H5/H6 for Cyt/Ura, and H1′ for all nucleotides), NOESY and TOCSY spectra were recorded using 100% D$_2$O samples at both 283 K and 298 K. The $^{13}$C chemical shifts from the nonexchangeable C-H vector were assigned by mapping proton assignments on $^{13}$C-$^1$H HSQC spectra individually recorded for the aromatic region (C8-H8, C6-H6 and C2-H2) and for the remaining vectors (C1′-H1′ and C5-H5). Protein-RNA binding experiments were all performed in NMR buffer of 50 mM sodium phosphate, pH 6.2, 150 mM KCl, 0.02% NaN$_3$, 5% D$_2$O.

For the chemical shift perturbation (CSP) analysis of RNA constructs, observed imino chemical shift values (H3 from Us and H1 from Gs) together with H2 of Ade from A-U base pairs were used to calculate the change in chemical shift Δ between the free and protein-bound states. For Site1, Site4 and Site5 constructs, additionally assigned nonexchangeable protons were included in the CSP analyses. Protein-bound RNA spectra of NOESY and TOCSY were assigned by tracing peak trajectories in the corresponding titration series, and only assignments of high confidence were included in the final analyses. Severely broadened and highly overlapping residues were excluded from the CSP calculations.

For the CSP analysis for Hexim1 proteins, the overall change in chemical shift Δ were calculated for protein $^{15}$N-$^1$H HSQC resonances between the free and RNA-bound states using the following equation:

$$\Delta = \sqrt{\Delta\delta_H{}^2 + 0.152 \times \Delta\delta_N{}^2}$$

where $\Delta\delta_H$ and $\Delta\delta_N$ are the differences between the $^1$H$_N$ and $^{15}$N chemical shifts of the two states being compared.

All the NMR spectra were collected at 298 K with an Avance Neo 800 MHz spectrometer equipped with TCI H&F cryoprobe, an Avance III HD 600 MHz spectrometer equipped with QCI HCNP cryoprobe or a Bruker DRX 500 MHz with cryoprobe. Data were collected with TopSpin (Bruker), processed with NMRPipe[74], and analyzed in NMRFAM-Sparky[75]. Hexim1 protein CSP values and PRE intensity ratios were plotted and visualized with Igor-Pro 9 software.

### Isothermal titration calorimetry

The dissociation constants (K$_D$) for binding between Hexim1 proteins and 7SK RNA constructs were determined using a MicroCal 200 ITC instrument (GE). RNA and protein were buffer exchanged using separate 3 kDa MWCO dialysis tubings into to the same bucket of ITC buffer (20 mM HEPES, pH 7.5, 150 mM KCl, 1 mM TCEP). Dilute dialyzed RNA solutions < 20 µM were heated at 95 °C for 5 min and snap cooled on ice for 1 hr right before the first set of titration experiments. RNA final concentrations were determined by UV absorbance at 90 °C at 260 nm with the Lambert-Beer law, using a Hewlett-Packard HP8453 diode-array UV/Visible spectrophotometer with Peltier temperature control. Extinction coefficients of melt RNA were calculated by the OligoAnalyzer Tool from Integrated DNA Technologies. RNA stock solutions were kept at room temperature to avoid cold-induced oligomerization for the 2-3 days duration of one complete titration set. RNA at concentration 63-195 µM in the syringe was titrated into 7−40 µM protein in the sample cell at 295 K, or for reversed titration order protein at concentrations of 102−120 µM in the syringe was titrated into 6−12 µM RNA in the sample cell at 295 K. Calorimetric data was fit using ORIGIN 7 (Micro-Cal). The binding parameters stoichiometry (N), entropy(ΔS), enthalpy (ΔH) and association constant (K$_A$) were kept as floating variables during each fit. ITC experiments were performed in

biological triplicate or quadruplicate ($n$ = 3 or 4) to verify reproducibility and estimate experimental error. No statistical methods were used to predetermine sample sizes, as $n$ = 3 is standard for quantitative biophysical measurements of purified systems. Full thermodynamics parameters and individual replicate numbers are listed in Supplementary Table 1.

### Multiple sequence alignment
Initial list of orthologs of Hexim1 and Hexim2 genes were extracted from NCBI gene cards (Gene ID: 10614 and 124790). Truncated and low-confidence genes were removed by manual inspection, resulting in 294 genes for Hexim1 and 210 genes for Hexim2. Sequence alignments were performed by COBALT (NCBI) and visualized using Jalview[76]. Five sequences for each Hexim1 and Hexim2 were selected for the final display, while the sequence placements were kept from the full alignments. A list of HIV-1 Tat protein sequences and alignment were acquired from UniProt, visualized using Jalview and one long isoform from each group and subtype of virus were selected for the final display.

### Reporting summary
Further information on research design is available in the Nature Portfolio Reporting Summary linked to this article.

## Data availability
The data supporting the findings of this study are available from the corresponding authors upon request. Raw data analyses are included as a combined supplementary data spreadsheet for protein CSPs, protein PRE intensity ratios, and RNA CSPs. Backbone chemical shift assignments of BR-L-AR protein, and backbone/sidechain chemical shift assignments of SL1-dI$_{\Delta U}$ RNA-bound BR-L-AR protein have been deposited in the Biological Magnetic Resonance Data Bank (BMRB), under accession IDs 53441 and 53442, respectively. Backbone and sidechain chemical shift assignments of additional monomeric protein constructs BR-L and BR have been deposited in BMRB under accession IDs 53446 and 53447, respectively. Proton chemical shift assignments of RNA constructs, SL1-dI (with additional carbon chemical shifts), SL1-dI$_{\Delta U}$, SL1-dII, SL1-d, SL1-dIIm, SL1-p, SL1-mp, SL1-pII and SL1alt$^{UUCG}$, have been deposited in BMRB under accession IDs 53443, 53444, 53445, 53448, 53449, 53450, 53451, 53452, 53453, 53454, respectively. Source data for the figures and Supplementary Figures are provided as a Source Data file. Source data are provided in this paper.

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

## Acknowledgements

This work was supported by the National Institutes of Health (NIH) R01AI155170 and R35GM131901 to J.F., American Heart Association Postdoctoral Fellowship 20POST35210850 to Y.Y. and American-Italian Cancer Foundation Postdoctoral Fellowship to M.G.M. The UCLA-DOE NMR core facility is supported in part by NIH instrumentation grants S10OD016336 and S10OD025073 and Department of Energy DE-FC02-02ER63421. We thank Catherine D. Eichhorn and Yanjiao Wang for helpful discussions.

## Author contributions

Y.Y., M.G.M., and J.F. conceptualized the study and designed the experiments. Y.Y. and M.G.M. performed the majority of the experiments and data analysis. S.G. prepared RNA and protein samples and acquired NMR, NativePAGE, and mass photometry data. Y.W. designed SL1 proximal constructs and performed NMR assignments. C.S., N.A., and X.W. helped with sample preparation for NMR and ITC experiments. Y.Y. and J.F. wrote the manuscript, and all authors participated in editing.

## Competing interests

The authors declare no competing interests.
