## [Transparent Peer Review file · Nature Communications]

HEXIM1 inter-monomer autoinhibition governs 7SK RNA binding specificity and P-TEFb inactivation

Corresponding Author: Professor Juli Feigon

Version 0:

Reviewer comments:

Reviewer #1

(Remarks to the Author)

The manuscript by Yang et al. describes a detailed NMR mapping and conformation analysis of the human protein Hexim1 targeting the small nuclear RNA 7SK. The 5' stem loop (SL1) of the 332 nt 7SK RNA interacts with its bulge regions with the basic regions of the homodimer Hexim1. Two different conformations have been proposed for 7SK SL1 (linear and circular) with different conformations and accessibilities of GAUC-containing bulge regions. Using sophisticated NMR-mapping experiments of a central construct of Hexim1, which is composed of a basic region, a linker region, and an acidic region (BR-L-AR, together 112 aa) or parts of it (BR-half, BR, BR-L), in combination with different constructs of the 7SK RNA, the authors identify two high affinity binding sites (1 and 2) and two lower affinity binding sites (4 and 5) in the linear conformation of SL1 as well as one high affinity binding site (3) in the circular conformation of SL1. The interactions to the three high affinity binding sites are validated by ITC experiments (Kds of 83 to 230 nM). Using the cysteine-reactive paramagnetic spin label MTSL in combination with site-directed mutagenesis for covalent attachment, the authors identify sites of extensive line broadening that give rise to conformational changes. They propose an autoinhibited masking of the central CDK9-inhibition motif PYNT by the flanking basic and acidic regions that might act in trans in the Hexim dimer assembly (Fig. 6).

Overall, this is a sound NMR study of a conformation and interaction analysis of Hexim1 with 7SK SL1 RNA. The manuscript is hard to follow for a non-NMR-specialist (particularly pages 9-11), making it a weaker candidate to the transcription regulation-interested broader community. Text and Figures, however, are well prepared. I have only a few remarks to this detailed and specific study.

Comments:

Ser158 in the basic region of Hexim1 has been described to be phosphorylated to regulate P-TEFb activity. The authors should characterize a BR-L-AR mutant where S158 is mutated to E (or even EE) to analyse conformational changes of the autoinhibited structure with a phosphorylation mimicry.

The C-terminal Cyclin T1/T2 coiled-coil binding domain (shown e.g. in the cartoon display of Fig. 1) has been determined by NMR spectroscopy including its chemical shift assignment. The N-terminal extension of this construct was shown to be feasible.

The linker region has been well analysed in previous studies (PMID 22588304) including a detailed characterization of the inhibitory potential of the PYNT motif in kinase activity experiments, showing that T205 (but not Y as e.g. in p27-Kip1 for CDK2 inhibition) is the main driver of kinase inhibition. The statement that "The central part of Hexim1 (150-250) has not been characterized previously, with the exception of a small region (~20 aa nuclear localization signal stretch) that binds 7SK RNA, also referred to as Basic Region (BR, aa 150-169)7,8." (Introduction section, page 3) seems therefore not correct giving previous work described in references 9 and 19.

Reviewer #2

(Remarks to the Author)

The 7SK RNA-protein complex (RNP) is an important factor for transcription regulation because it sequesters P-TEFb the kinase complex (CDK9 and Cyclin T1) that is required for transcription elongation after RNA polymerase II pausing. There were several 7SK states previously detected in cells but for P-TEFb regulation a core complex consisting of 7SK RNA, HEXIM1 (or 2), MePCE and LARP7 seems to be crucial. Here, Yang, Feigon and colleagues try to define how HEXIM1 contacts 7SK RNA and how RNA binding influences its conformation. The authors generated a large number of truncated 7SK RNA sections and several HEXIM1 truncation constructs which they use for NMR and binding assays (EMSA, ITC).

The paper is terribly confusing and very hard to read. Some of this is due to different sets of protein constructs and RNA designs being used for different assays, but also the text jumps back and forth between figures, does not explain the relevance of NMR findings in general terms, and for several RNA constructs it is difficult to find out what they correspond to. It is often unclear, what conclusions are based on because the presented data does not warrant the conclusions. It is most problematic, that some assays (NMR, EMSA, ITC) were carried out with one set of RNA and protein construct, while others were only done with different sets. But then conclusions from the first set are simply "carried over" to the second set of constructs. Furthermore, quite ambivalent PRE data is overinterpreted to fit into an autoinhibition model. In its current form, I do not see how the conclusions that were made are supported by experimental evidence. To change this issue, minimally EMSA and ITC binding assays as well as NMR titration experiments would have to be carried out for corresponding and meaningful sets of protein/RNA constructs. Also complementary evidence to the current PRE data is required to make the autoinhibition model convincing.

In addition, the artificial nature of the shortened RNA and protein constructs, as well as the minimal set of relevant components used in the assays (only part of 7SK RNA and HEXIM1) makes it unclear how relevant the study is to understand what happens in cells and what any of the findings tell us of how 7SK-HEXIM1 binding to P-TEFb could be regulated.

In summary, before publication can be considered this manuscript needs severe restructuring and streamlining of text and figures, plus significant experimental work.

Below, I am listing some of the issues in detail in the hopes that it helps to improve the manuscript.

RNA and protein constructs:

- * It is often unclear what the RNA constructs used in each experiment exactly look like, these should be shown as inserts in each assay or minimally the complete set should be shown in Fig.3A.
- * Why are deltaU constructs used? From the EMSA studies it seems that SL1-dldeltaU binds much less well to HEXIM1 than SL1-dI (compare Suppl.Fig.5B gel1 and gel2). Yet, in ITC experiments only SL1-dI is used and not the corresponding deltaU RNA. But then deltaU constructs are used for NMR. These are completely artificial constructs, why are they relevant?
- * Why are different sets of RNAs used for EMSA vs. ITC vs. NMR? Why are different HEXIM1 constructs used for EMSA vs. ITC vs. NMR experiments? I understand that solubility is an issue for some protein constructs, which makes it difficult to use them for NMR but for ITC and EMSA this should not be a problem, given that RNA affinities are in the nano to micro molar range. These differences make the manuscript not only confusing and difficult to follow, they also draw comparisons and conclusions into question.

RNA NMR experiments and CSPs:

- * p.8 states "Using chemical shift perturbation (CSP) of RNA imino proton resonances in 1H-1H NOESY spectra in the absence versus presence of monomeric Hexim1 BR-L-AR or BR, we identified SL1 Sites 1 and 2 as two high affinity binding sites, and Sites 4 and 5 as two lower affinity binding sites (Fig. 3 and Supplementary Fig. 7)."
- > Several problems here: # How can CSPs inform on high/low affinity binding sites?
- # CSPs for RNA is never shown anywhere, only (difficult to distinguish) shades of blue in the base letters (Fig.3A, S5A) indicate them to some extent. CSPs should be shown in a graph analogous to protein CSPs.
- # NMR experiments are done with inconsistent protein constructs, which on top differ from what the sentence stated i.e. SL-dldeltaU with BR-L-AR, SL1-dII with BR-L, SL1-mdII with BRhalf. This means, BR is never used, instead BR-L and BRhalf were used but only for one RNA each. If both protein and RNA are changed in each experiment, how can conclusions about affinities to individual RNA binding sites be made?

- * p.8 states "when using longer 7SK RNA constructs that include two adjacent binding sites, the higher affinity site of the two is occupied by stoichiometric addition of Hexim1 BR as shown by NMR CSP of RNA iminos, i.e. in SL-d Site 1 shows CSP instead of Site 4, in SL1-mdII Site 2 shows CSP instead of Site 4, and in SL1-pm Site 2 shows CSP instead of Site 5 (Fig. 3A and Supplementary Fig. 5A);"
- > Where is this data? There is no SL1-d NMR dataset shown anywhere. Furthermore, binding site 4 in SL1-d in the scheme is shown in grey, which means "not assigned". So how can any conclusion about protein binding to this region be made?

EMSAs/ITC - Several issues with presentation, analysis and interpretation of EMSAs and ITC experiments:

- * Instead of reporting RNA-protein ratios, the EMSA binding assays should rather report concentrations. Kd values should be calculated and compared to ITC data.
- * It is unclear how the assignments on the side of the EMSA gels with binding sites 1/2/4/5 can be made, especially for SL1-ddeltaU, when both site 4 and site 1 were bound separately at roughly 0.9 RNA/protein ratio (compare Suppl.Fig.5B gel1 and gel3). All that can be concluded is that the first shift has probably 1 HEXIM1 bound and the second shift has probably 2 HEXIM1s bound to the RNA.
- * Similarly, statements in the text are not supported by the data. E.g. p.9 "Indeed, EMSAs of SL1 constructs with full-length

Hexim1 show that Hexim1 dimer binds first to Sites 1 and 2 and subsequently to Sites 4 and 5 (Supplementary Fig. 5D,E)." All that can be concluded from this EMSA is that two proteins bind at 1.2 ratio and another two at 2.4 ratio. Which RNA sites they bind to cannot be seen in this assay.

*All ITC curves should be included in the supplement. Some of the listed affinities are comparably weak (micro molar Kd) and fitting gives substoichiometry N-values, which could indicate that these curves do not reliably report N or Kd any more and should rather be given as approximate boundary values (rather than four significant digits).

PRE experiments and PYNT autoinhibition model:

* In Fig. 4B-E regions with PRE intensity ratios below mean-0.5sigma and mean-1 sigma are supposedly shaded in orange and red. Yet in almost every one of the shaded regions several residues lie above the supposed cut-off (or are simply not assigned) and conversely in the white regions several residues have lower than cut-off ratios (or are simply not assigned). Thus, the shaded areas give an oversimplified and scewed picture of the data and this is cemented further by the structure cartoons presented on top. Based on the data one could equally well argue that the AR is in transient contact with any one of the other regions of HEXIM1. Additional and complementary evidence would be required to make this section convincing.

* On top of the weak data, for these experiments it is especially relevant that they are carried out in the context of a truncated HEXIM1 that cannot form the obligate dimer. The N-terminal HEXIM1 regions could equally well bind back to the coiled-coil section, which would also lead to reduced RNA affinity of full length HEXIM1 for 7SK RNA (as determined by ITC).

* If the PYNT motif is truly involved in reducing RNA affinity and HEXIM1 making binding "more selective", then a deletion of this short motif should increase RNA affinity of HEXIM1 and make it "less selective".

* Conceptually the model also poses a problem as most of HEXIM1 in cells is bound to dsRNA (Li et al. NAR 2007), and should therefore be relieved of the autoinhibited conformation and associate with P-TEFb according to the proposed model, but this seems not to be the case in cells. Furthermore, in previous studies the very N-terminus that is not studied here (aa 1-120) has an auto-inhibitory effect (Li et al. JBC 2005).

Figure citations - The text jumps back and forth between figures some in the wrong order and some figure citations seem to be entirely wrong e.g. Fig. 2B should be cited on p.5 ("...referred to as BR-L-AR in the following text.") before Fig. 2A is coming in. There is no Fig.3E but it is referred to in the text (p.7 "...bulge (U40U41) is also present 3' to the alternative GAUC palindrome (Fig. 3E).")

Chemical exchange - If this paper is meant for a general audience, the authors should explain what slow, intermediate and fast exchange mean in context of the system studied and the complexes that are formed here (e.g. on p.6, p.8).

Abstract final sentence "These results provide mechanistic insights into Hexim-RNA specificity and explain how P-TEFb can be effectively regulated to respond to changing levels of transcriptional signaling." - This conclusion is not warranted given that the authors have not carried out any assays in cells and used only artificial in vitro systems with few (and mostly severely truncated) components.

Figure inconsistencies - I understand that some peaks are broadened upon addition of RNA but it still looks like some bars are missing in Suppl.Fig.4C basic region in comparison to Fig. 2D/E? Also the baseline at 0 is missing in Suppl.Fig.4C.

Suppl. Fig. 1 and 2: The sequence alignments seem to be cut out of larger alignments because they display numerous empty columns that hold no information. These empty columns should be removed to make the alignments more compact and easier to understand.

Version 1:

Reviewer comments:

Reviewer #1

(Remarks to the Author)

The manuscript has been significantly improved, although it remains rather complicated to read in places. However, this is probably unavoidable when it comes to the description of NMR chemical shift perturbation data and the explanation of MTSL spin labelling experiments. The use of RNA constructs is a bit better to follow now and the confusion with mixed constructs in neighbouring panels is partly resolved. The discussion benefits a lot from the new model figure (Fig. 8) of the proposed regulation of P-TEFb by the 7SK snRNP in the main text. The figures are well prepared, although they may be difficult to follow for non-NMR specialists.

I have no further comments to this expert study.

Reviewer #3

(Remarks to the Author)

A few points to consider revising:

Review:

- Constructs were clarified reasonable, paper still hard to read

- Fig. 1E: Where do percentages come from? How are the actual Mass-Photometry size correlated to the thought to be observed species... not clear

- According to native gel (Fig. S4a), Hexim1 can also form tetramers. not indicated, please clarify

- The other species in the mass photometry studies are ignored? Why, please explain
- o Fig. S4c/1E the calculation of trimer and tetramer are not evident
- o Which of the data for BR-L-AR is for 40 nM or 4-25 μ M?
- CSP mean doesn't make sense – one usually establishes CSP baseline that is not changing, and anything bigger than the std of this is considered a change in CSP... so with mean you are already excluding some relevant CSPs.

Reviewer #1 (Remarks to the Author):

The manuscript by Yang et al. describes a detailed NMR mapping and conformation analysis of the human protein Hexim1 targeting the small nuclear RNA 7SK. The 5' stem loop (SL1) of the 332 nt 7SK RNA interacts with its bulge regions with the basic regions of the homodimer Hexim1. Two different conformations have been proposed for 7SK SL1 (linear and circular) with different conformations and accessibilities of GAUC-containing bulge regions. Using sophisticated NMR-mapping experiments of a central construct of Hexim1, which is composed of a basic region, a linker region, and an acidic region (BR-L-AR, together 112 aa) or parts of it (BR-half, BR, BR-L), in combination with different constructs of the 7SK RNA, the authors identify two high affinity binding sites (1 and 2) and two lower affinity binding sites (4 and 5) in the linear conformation of SL1 as well as one high affinity binding site (3) in the circular conformation of SL1. The interactions to the three high affinity binding sites are validated by ITC experiments (Kds of 83 to 230 nM). Using the cysteine-reactive paramagnetic spin label MTSL in combination with site-directed mutagenesis for covalent attachment, the authors identify sites of extensive line broadening that give rise to conformational changes. They propose an autoinhibited masking of the central CDK9-inhibition motif PYNT by the flanking basic and acidic regions that might act in trans in the Hexim dimer assembly (Fig. 6).

Overall, this a sound NMR study of a conformation and interaction analysis of Hexim1 with 7SK SL1 RNA. The manuscript is hard to follow for a non-NMR-specialist (particularly pages 9-11), making it a weaker candidate to the transcription regulation-interested broader community. Text and Figures, however, are well prepared. I have only a few remarks to this detailed and specific study.

Response:

We thank the reviewer for the positive comments and the concern for the suboptimal readability towards non-NMR specialist readers. To address this important concern, we have significantly rewritten and rearranged the manuscript, including the NMR sections, and incorporated new and enlarged cartoons to help with visual interpretation of the NMR results (revised Figs. 3 and 4). We also now have a new section on S158 phosphorylation and substitution that should increase the interest of the broader transcription community.

Comments:

Ser158 in the basic region of Hexim1 has been described to be phosphorylated to regulate P-TEFb activity. The authors should characterize a BR-L-AR mutant where S158 is mutated to E (or even EE) to analyse conformational changes of the autoinhibited structure with a phosphorylation mimicry.

Response:

We thank the reviewer for this great suggestion. To this end, we performed two additional sets of experiments: (1) in vitro phosphorylation of BR-L-AR by PKA catalytic subunit, which showed S158 phosphorylation (assayed by NMR) had minimal impact on the autoinhibited conformation, and (2) ITC binding measurements of a NBLA-S158EE mutant binding to 7SK, which showed significantly reduced RNA binding. Together, these observations suggest that in vivo S158 phosphorylation leads to reduced RNA binding without significantly disrupting Hexim autoinhibition, thereby triggering release of P-TEFb as a net result. See the new Results section titled "*S158 phosphorylation minimally affects autoinhibition and S158EE impairs RNA binding*" (Lines 289–312), new Fig. 6 and Supplementary Fig. 23, of the revised manuscript, for details.

The C-terminal Cyclin T1/T2 coiled-coil binding domain (shown e.g. in the cartoon display of Fig. 1) has been determined by NMR spectroscopy including its chemical shift assignment. The N-terminal extension of this construct was shown to be feasible.

Response:

We were aware of the excellent papers on the Hexim coiled-coil domain from the Geyer lab, and apologize for neglecting to reference the NMR structure in our previous manuscript. Regarding N-terminal extension of coiled-coil construct, we could indeed express and purify soluble proteins of BR-L-AR-CC and L-AR-CC constructs. Unfortunately, BR-L-AR-CC construct exhibited severe line broadening for the entire BR-L-AR region by NMR, similar to full-length Hexim1, despite our efforts in optimizing temperature, salt, and deuterium labeling of the constructs. We therefore included new nativePAGE and mass photometry data including Hexim1 dimer and L-AR-CC dimer in our revised manuscript (Lines 109–126, Fig. 1 and Supplementary Fig. 4), which provide strong evidence of stable inter-monomer interactions.

The linker region has been well analysed in previous studies (PMID 22588304) including a detailed characterization of the inhibitory potential of the PYNT motif in kinase activity experiments, showing that T205 (but not Y as e.g. in p27-Kip1 for CDK2 inhibition) is the main driver of kinase inhibition. The statement that "The central part of Hexim1 (150-250) has not been characterized previously, with the exception of a small region (~20 aa nuclear localization

signal stretch) that binds 7SK RNA, also referred to as Basic Region (BR, aa 150-169)7,8.” (Introduction section, page 3) seems therefore not correct giving previous work described in references 9 and 19.

Response:

We thank the reviewer for this correction. We meant “has not been structurally characterized”. We have clarified this in the revised manuscript. We have also incorporated the important details related to linker region analysis in PMID 22588304 in the revised manuscript’s Introduction section (Lines 62).

Reviewer #2 (Remarks to the Author):

The 7SK RNA-protein complex (RNP) is an important factor for transcription regulation because it sequesters P-TEFb the kinase complex (CDK9 and Cyclin T1) that is required for transcription elongation after RNA polymerase II pausing. There were several 7SK states previously detected in cells but for P-TEFb regulation a core complex consisting of 7SK RNA, HEXIM1 (or 2), MePCE and LARP7 seems to be crucial. Here, Yang, Feigon and colleagues try to define how HEXIM1 contacts 7SK RNA and how RNA binding influences its conformation. The authors generated a large number of truncated 7SK RNA sections and several HEXIM1 truncation constructs which they use for NMR and binding assays (EMSA, ITC).

The paper is terribly confusing and very hard to read. Some of this is due to different sets of protein constructs and RNA designs being used for different assays, but also the text jumps back and forth between figures, does not explain the relevance of NMR findings in general terms, and for several RNA constructs it is difficult to find out what they correspond to. It is often unclear, what conclusions are based on because the presented data does not warrant the conclusions.

Response:

We thank the reviewer for pointing out the structural issues with our manuscript, as we had indeed a large body of inter-connected datasets to describe. We have significantly re-organized the Results sections, regrouped relevant protein and RNA construct designs in the main and supplementary figures, and now clearly lay out what was learned at the end of each section. We hope this along with additional new experiments will convince the reviewer that our data does indeed warrant our conclusions.

It is most problematic, that some assays (NMR, EMSA, ITC) were carried out with one set of RNA and protein construct, while others were only done with different sets. But then conclusions from the first set are simply “carried over” to the second set of constructs. Furthermore, quite ambivalent PRE data is overinterpreted to fit into an autoinhibition model. In its current form, I do not see how the conclusions that were made are supported by experimental evidence. To change this issue, minimally EMSA and ITC binding assays as well as NMR titration experiments would have to be carried out for corresponding and meaningful sets of protein/RNA constructs. Also complementary evidence to the current PRE data is required to make the autoinhibition model convincing.

Response:

We have rewritten the Results to address the confusions regarding different constructs, and added comparisons between constructs whenever possible. In addition, we have now included nativePAGE and mass photometry results to provide complementary analysis for the PRE results, and acquired new evidence for inter-monomer interactions mediated by the central region for both NMR-range micromolar concentration (nativePAGE) and dilute nM concentration (mass photometry). These results have been incorporated into the revised manuscript (Lines 109–126, Fig. 1 and Supplementary Fig. 4).

In addition, the artificial nature of the shortened RNA and protein constructs, as well as the minimal set of relevant components used in the assays (only part of 7SK RNA and HEXIM1) makes it unclear how relevant the study is to understand what happens in cells and what any of the findings tell us of how 7SK-HEXIM1 binding to P-TEFb could be regulated.

Response:

First, we already know that the two 7SK RNA conformations and a minimum Site1 were relevant both in vitro and in cells from the body of literature, especially from our recent structure/function work (Yang et al. *Mol Cell* 2022) and DANCE-MaP ensemble-deconvoluted chemical probing work (Olson et al. *Mol Cell* 2022), which both illustrated the correlation between 7SK RNA conformations and Hexim1/P-TEFb sequestration/release in cells. Thus, studying the site-specific difference between linear and circular 7SK in binding Hexim is key to understanding the eukaryotic transcription regulation by 7SK RNP.

Second, it is standard to use divide-and-conquer strategy to dissect RNA binding sites and compare to larger RNA and protein, since many RNA (and DNA)-binding proteins exhibit both specific and non-specific binding modes. Since previous structural and functional studies on Hexim–7SK binding focused exclusively on linear 7SK Site1 and on BR_{ARM1} or BR of Hexim1, and the fact that Hexim was proposed as a non-specific dsRNA binder in one study, *our study here is the first that maps the complete picture of the Hexim–7SK RNA interaction landscape*. Using a divide-and-conquer strategy to compare short to long RNA, and compare binding of Hexim1 monomer constructs [BR_{ARM1}, BR, BR-L, BR-L-AR, NBLA] to full-length Hexim1 dimer, was key to distinguish specific (high-affinity) versus non-specific (low-affinity) RNA binding sites by monomeric Hexim1 constructs and to compare them to binding by the autoinhibited full-length Hexim1 dimer. We could not have understood or explained how binding of full-length Hexim1 to dual sites on 7SK SL1 regulated autoinhibition and therefore P-TEFb binding and inactivation without these experiments.

We also note that a previous study looked at HIV-1 Tat binding to 7SK SL4 on the basis of which it was concluded that Hexim1 also binds SL4 (Durney et al. JMB 2010 PMID: 20816986). However, we now know from our structural and functional work (Yang et al. Mol Cell 2022) that SL4 is unavailable for Hexim1 binding in the correctly assembled 7SK core RNP, since its major groove is fully occupied by Larp7–SL4 interaction. This highlights the need to look at the physiologically relevant Hexim1 binding site on 7SK RNA, in order to avoid arriving at the wrong conclusion about Hexim1 binding specificity, 7SK RNP assembly, or Hexim–Tat competition for 7SK. In fact, we can now confidently say that SL4 (even in the absence of Larp7) is not a specific site for Hexim1 based on our current study comparing RNA binding sites. SL4 contains a dynamic C–C–G base triple from a CU bulge at the 3' side of the proximal stem, failing to meet the two requirements we determined for Hexim1 high-affinity binding sites: U–A–U base triple from a U-rich bulge at the 5' side of the A–U rich stem.

In summary, before publication can be considered this manuscript needs severe restructuring and streamlining of text and figures, plus significant experimental work.

Below, I am listing some of the issues in detail in the hopes that it helps to improve the manuscript.

We thank the reviewer for taking the time to point out perceived issues with our data and analysis. There is indeed a wealth of data in this manuscript. We hope that with our revisions the significance of our results can now be properly appreciated.

RNA and protein constructs:

* It is often unclear what the RNA constructs used in each experiment exactly look like, these should be shown as inserts in each assay or minimally the complete set should be shown in Fig.3A.

Response:

We now show a complete set of RNA constructs used for RNA NMR in Fig 5 (previous Fig. 3A), and also added RNA constructs (together with NMR CSP mapping when available) to all relevant Supplementary figures (6,8,9,14,16–22,24,25).

* Why are deltaU constructs used? From the EMSA studies it seems that SL1-d-deltaU binds much less well to HEXIM1 than SL1-dl (compare Suppl.Fig.5B gel1 and gel2). Yet, in ITC experiments only SL1-dl is used and not the corresponding deltaU RNA. But then deltaU constructs are used for NMR. These are completely artificial constructs, why are they relevant?

Response:

The reviewer is correct that SLI-deltaU binds less well to Hexim1 than SL1-dl. We previously chose the deltaU construct to help with NMR assignments, since the wt-version (SL1-dl) causes even more severe line broadening for both protein and RNA spectra. We have now included data from SL1-dl titrating with protein in Fig. 2 and Supplementary Fig. 6, and showed that the same Site1 is used for binding (Fig. 2C). We have also incorporated an interesting comparison between SL1-dl-deltaU and SL1-dl CSP results (Fig. 2C and compare Supplementary Figs. 7 and 11), which explains in a nucleotide-specific manner why a previous study observed a contribution of U63 bulge to Hexim1 binding (Martinez-Zapien et al. NAR 2017), and also explains why Site1 is better than all the other sites we looked at. Please see Results section “Characterization of Hexim monomer–7SK RNA Site1 interaction” (Lines 141–166) for details.

*Why are different sets or RNAs used for EMSA vs. ITC vs. NMR? Why are different HEXIM1 constructs used for EMSA vs. ITC vs. NMR experiments? I understand that solubility is an issue for some protein constructs, which makes it difficult to use them for NMR but for ITC and EMSA this should not be a problem, given that RNA affinities

are in the nano to micro molar range. These differences make the manuscript not only confusing and difficult to follow, they also draw comparisons and conclusions into question.

Response:

We used the same constructs for EMSA and NMR, in order to confirm binding stoichiometry. There are three longer RNA constructs used for NMR, as a part of the suite for divide-and-conquer strategy. The RNA constructs used are now shown with the ITC data, which clearly shows that the same RNAs are used for NMR and ITC experiments (Supplementary Fig. 24). (We note that SL1-d(api), a construct studied previously by NMR (Pham et al. *Commun Biol* 2022), is essentially the same as SL1-d, i.e. differs only in the apical loop.)

RNA NMR experiments and CSPs:

*p.8 states "Using chemical shift perturbation (CSP) of RNA imino proton resonances in 1H-1H NOESY spectra in the absence versus presence of monomeric Hexim1 BR-L-AR or BR, we identified SL1 Sites 1 and 2 as two high affinity binding sites, and Sites 4 and 5 as two lower affinity binding sites (Fig. 3 and Supplementary Fig. 7)."

-> Several problems here: # How can CSPs inform on high/low affinity binding sites?

Response:

For the NMR studies, RNA binding sites were established using the shorter RNAs. Then for the longer RNAs that contained more than one binding site, titration of the Hexim1 constructs shows clearly which site is occupied at 1:1 ratio (i.e. has higher affinity) based on the observed CSP. We have re-organized the relevant figure panels and added small cartoons (now Fig. 5A) to better illustrate these direct comparisons. The binding site relative affinities determined by NMR are interpreted in conjunction with EMSA stoichiometric shifts (Supplementary Fig. 14). Exchange behaviors on NMR titration timescale are also consistent with the relative order of affinities, where lower affinity binding shifts to faster exchange likely due to increased off-rate.

CSPs for RNA is never shown anywhere, only (difficult to distinguish) shades of blue in the base letters (Fig.3A, S5A) indicate them to some extent. CSPs should be shown in a graph analogous to protein CSPs.

Response:

The CSPs are shown on the secondary structures of the RNAs to make it easier for the reader to see where the binding sites are. We now include individual Supplementary Figures for each RNA construct, their imino walk assignments, CSP plots, and 1D titration series (main Figs. 2 and 5, Supplementary Figs. 6, 8, 16–22).

NMR experiments are done with inconsistent protein constructs, which on top differ from what the sentence stated i.e. SL-dIdelaU with BR-L-AR, SL1-dII with BR-L, SL1-mdII with BRhalf. This means, BR is never used, instead BR-L and BRhalf were used but only for one RNA each. If both protein and RNA are changed in each experiment, how can conclusions about affinities to individual RNA binding sites be made?

Response:

We now show by direct comparison among BR-L-AR, BR-L, BR and BR_{ARM1} (renamed from BR_{half} for clarity) constructs that the same ARM1 amino acids are perturbed by binding to Site1 RNA (main Fig. 2 and Supplementary Fig. 10) for all these protein constructs. For longer constructs of RNA, BR_{ARM1} is used to minimize line-broadening. A direct comparison of RNA CSP between BR-L-AR and BR_{ARM1} binding to Site1 nucleotides is shown in Supplementary Fig. 17D. In the longer RNA constructs with two or more sites, conclusions of relative order of binding affinity were drawn from each stand-alone titration experiment, not by comparisons across different RNA constructs (revised Fig. 5A). Quantitative comparison of binding affinities was done with ITC measurements, where the experimental designs were informed by the NMR and EMSA experiments so that we know what binding event was being observed in each ITC titration.

*p.8 states "when using longer 7SK RNA constructs that include two adjacent binding sites, the higher affinity site of the two is occupied by stoichiometric addition of Hexim1 BR as shown by NMR CSP of RNA iminos, i.e. in SL-d Site 1 shows CSP instead of Site 4, in SL1-mdII Site 2 shows CSP instead of Site 4, and in SL1-pm Site 2 shows CSP instead of Site 5 (Fig. 3A and Supplementary Fig. 5A);"

-> Where is this data? There is no SL1-d NMR dataset shown anywhere. Furthermore, binding site 4 in SL1-d in the scheme is shown in grey, which means "not assigned". So how can any conclusion about protein binding to this region be made?

Response:

We now report non-exchangeable base and H1' proton assignments for shorter RNA constructs, and provide evidence that the imino proton CSPs can capture the Hexim1 binding sites very well, when compared to CSPs from

the whole set of proton assignments (Fig. 2B, filled signs versus empty signs, and Supplementary Fig. 8C).

Regarding SL1-d NMR data, as described 2 points above, we now include individual Supplementary Figures for each RNA construct, their imino walk assignments, CSP plots, and 1D titration series (SL1-d data shown in main Fig. 5, Supplementary Figs. 17). Regarding the gray residues, we have obtained additional assignments for imino and base protons. We note that gray indicates not available CSP pairs, due to line broadening of bound spectrum or spectral overlap (now clarified in figure legends).

EMSAs/ITC - Several issues with presentation, analysis and interpretation of EMSAs and ITC experiments:

*Instead of reporting RNA-protein ratios, the EMSA binding assays should rather report concentrations. K_D values should be calculated and compared to ITC data.

Response:

The concentrations used for EMSA were chosen to determine *stoichiometry* in conjunction with the NMR binding experiments, not binding affinity, hence molar ratios were used. Therefore these EMSAs can only be analyzed for K_D apparent. For the reviewer, we fit the EMSA bands to $K_{D,app}$ values, also called microscopic dissociation constants, to assist in comparison across EMSA results. We have added "Notes for EMSA binding curve fitting" below Supplementary Fig. 14, which provide details for the calculations and Supplementary Fig. 15 for fitting results.

*It is unclear how the assignments on the side of the EMSA gels with binding sites 1/2/4/5 can be made, especially for SL1-deltaU, when both site 4 and site 1 were bound separately at roughly 0.9 RNA/protein ratio (compare Suppl.Fig.5B gel1 and gel3). All that can be concluded is that the first shift has probably 1 HEXIM1 bound and the second shift has probably 2 HEXIM1s bound to the RNA.

*Similarly, statements in the text are not supported by the data. E.g. p.9 "Indeed, EMSAs of SL1 constructs with full-length Hexim1 show that Hexim1 dimer binds first to Sites 1 and 2 and subsequently to Sites 4 and 5 (Supplementary Fig. 5D,E)." All that can be concluded from this EMSA is that two proteins bind at 1.2 ratio and another two at 2.4 ratio. Which RNA sites they bind to cannot be seen in this assay.

Response:

It is true that sites cannot be determined from EMSA alone. However, as explained above, EMSA results were used in conjunction with NMR titration experiments to determine which site (high-affinity) is occupied at 1-to-1 stoichiometry but not the other site (lower-affinity) out of two. We have now added NMR CSPs mapped on RNA constructs next to each EMSA gel, and fit EMSA bands to $K_{D,app}$ to better illustrate these analyses. In addition, we have obtained quantitative K_D by ITC, as addressed below.

*All ITC curves should be included in the supplement. Some of the listed affinities are comparably weak (micro molar K_D) and fitting gives substoichiometry N-values, which could indicate that these curves do not reliably report N or K_D any more and should rather be given as approximate boundary values (rather than four significant digits).

Response:

We have now included all ITC curves in Supplementary Fig. 24. Direct ITC binding experiments can reliably determine K_D within the range of 100 μ M to 1 nM, i.e. our K_D values fell well within this range (2.1 μ M to 73 nM). For ITC experiments, a 1-to-1 stoichiometric binding often corresponds to a N value smaller than 1.0 ("substoichiometry"), e.g. the Calcium-EDTA control titration yields an N value of 0.7–0.8 on ITC. Since we did use the N values to compare one-site to two-site binding, and to illustrate impaired release of Hexim1 autoinhibition, we also did the appropriate control by doing the reverse ITC titrations by swapping protein from cell to syringe (less optimal due to more protein aggregation at higher concentrations). These experiments gave N values that are approximately 1/N of the equivalent titration pair in the original titration order. For example, SL1-dI was N=0.76, 1 over this N will be $1/0.76 = 1.32$, and reverse order SL1-dI had N=1.12 (slightly smaller than 1.32, as anticipated from more protein aggregation). We have now also cross-validated the accuracy of these N values by repeating key titrations with a newly acquired ITC instrument from a different company (TA instrument, Affinity ITC), and acquired the same N numbers as the previous instrument we used (Malvern, MicroCal ITC).

Regarding the reported values and errors, they are average and standard deviations from three or more independent ITC titration experiments, not just fitting error from one experiment (which is sometimes reported as boundary values). We have included this clarification in Fig. 7 and Supplementary Fig. 24 captions. The decimal places are kept identical, in order to assist fast comparison between weak and strong binding sites, but we have rounded up the weaker binders to three significant figures to reflect the three significant figures of ITC instrument reported values.

PRE experiments and PYNT autoinhibition model:

* In Fig. 4B-E regions with PRE intensity ratios below mean-0.5sigma and mean-1sigma are supposedly shaded in

orange and red. Yet in almost every one of the shaded regions several residues lie above the supposed cut-off (or are simply not assigned) and conversely in the white regions several residues have lower than cut-off ratios (or are simply not assigned). Thus, the shaded areas give an oversimplified and skewed picture of the data and this is cemented further by the structure cartoons presented on top. Based on the data one could equally well argue that the AR is in transient contact with any one of the other regions of HEXIM1. Additional and complementary evidence would be required to make this section convincing.

Response:

We have clarified in the text how the regions of mean-0.5sigma and mean-1sigma PREs were defined, i.e. these correspond to stretches with three or more non-contiguous residues, in order to assist with identifying which structural elements are involved. We have revised and enlarged the cartoons to better illustrate the PRE interpretation.

For *complementary evidence of autoinhibition*, we have added ITC data on AR and PYNT mutants (A3m, A12m and DeltaPYNT) in full-length Hexim1 dimer, to further support our model of inter-monomer autoinhibition. *These mutants all turned full-length Hexim1 into a non-specific RNA binder*, illustrating their significant roles in autoinhibition and RNA binding (Lines 364–370, Fig. 7 and Supplementary Fig. 25). We also obtained NativePAGE and mass photometry data that provide further support for the AR–BR centric inter-monomer interactions (Lines 109–126, Fig. 1 and Supplementary Fig. 4).

* On top of the weak data, for these experiments it is especially relevant that they are carried out in the context of a truncated HEXIM1 that cannot form the obligate dimer. The N-terminal HEXIM1 regions could equally well bind back to the coiled-coil section, which would also lead to reduced RNA affinity of full length HEXIM1 for 7SK RNA (as determined by ITC).

Response:

We cannot completely rule out a potential weak interaction between Hexim1 N-term and coiled-coil domain (also see discussion of autoinhibitory N-terminus below), but we have acquired new mass photometry data on mixed complexes between L-AR-CC and BR-L-AR, and between Hexim1 and BR-L-AR, which are consistent with BR-L-AR centric inter-monomer interactions. The fact that L-AR-CC construct does not form higher order complex at 1:4 molar ratio rules out any stable interaction between the coiled-coil domain and BR-L-AR at 40nM concentration.

* If the PYNT motif is truly involved in reducing RNA affinity and HEXIM1 making binding "more selective", then a deletion of this short motif should increase RNA affinity of HEXIM1 and make it "less selective".

Please see our response (2 points above) where we describe new AR and PYNT mutants (A3m, A12m and DeltaPYNT) that turn Hexim1 into non-specific binder.

* Conceptually the model also poses a problem as most of HEXIM1 in cells is bound to dsRNA (Li et al. NAR 2007), and should therefore be relieved of the autoinhibited conformation and associate with P-TEFb according to the proposed model, but this seems not to be the case in cells. Furthermore, in previous studies the very N-terminus that is not studied here (aa 1-120) has an auto-inhibitory effect (Li et al. JBC 2005).

Response:

Our NBLA construct spans residues 1–253, including the very N-terminus. Therefore, *any difference observed between NBLA monomer and Hexim1 dimer by ITC, is only due to the addition of coiled-coil dimerization domain, not due to a lack of N-terminus*. That said, our model does not exclude the auto-inhibitory effect of the N-terminus aa 1-120, but we cannot identify the responsible N-terminal elements, if any, by looking at the sequence alignment of Hexim1 and Hexim2 orthologs. For human Hexim1 sequence, its extreme N-terminus contains a slightly acidic and hydrophobic stretch (residues 1-26, net -4 charge, or -8 charge for residues 1-66), but the acidic residues are not clustered, rather this segment appears more amphipathic than acidic. In comparison, central region AR (-13 total net charges) contains clustered net -6 charges for n-AR, and clustered net -10 charges for c-AR. It was also interesting that in Li et al. JBC 2005, the DeltaN construct could still respond to 7SK RNA binding, which significantly enhanced its P-TEFb-inhibitory activity, consistent with releasing BR-L-AR mediated autoinhibition. We note in our revised manuscript that the inclusion of N-terminus increased the solubility of RNP during ITC titrations, pointing to a contribution of the disordered N-terminus to an altered protein biochemical property. It is possible that the extreme N-terminus could provide further stabilization to the BR–PYNT–AR interaction of the central region or contact CC domain. Due to length constraint, we added these discussions to “Supplementary discussion regarding N-terminus of Hexim and Tat” below Supplementary Fig. 26.

Please see our response (3 points above) where we describe new AR and PYNT mutants (A3m, A12m and DeltaPYNT) that turn Hexim1 into non-specific binder.

Regarding Hexim bound to dsRNA in cells, from our comparison of the five mapped RNA sites, we conclude that the determining factors for high-affinity Hexim binding site are an A-U rich stem with at least one flanking U-rich bulge at the 5' side, which will inform future studies of Hexim interactions with other cellular targets. Please see the refined model figure (Fig. 8) and Discussion third paragraph (Lines 402–414) for illustration and description of the issue of general Hexim1–RNA specificity. We would like to also mention a recently study published during our revision process (Graham et al. *Mol Cell* 2025) for live cell fluorescent imaging of 7SK RNP proteins, which showed that the portion of Hexim1 not bound to 7SK RNP exhibited a fast-diffusing peak (6-7 $\mu\text{m}^2/\text{s}$ versus 2-3 $\mu\text{m}^2/\text{s}$ peak for CycT1- and Larp7-bound peak), consistent with our “facilitated search” model, which implies that the pool of 7SK-free Hexim1 likely does not stably associate with other cellular dsRNA targets (unless these dsRNA targets contain a similar high-affinity site to what we determined for 7SK). We note that Li et al. *NAR* 2007 showed a successful enrichment of 7SK, not U2 and U6 RNA, through affinity pull down of Hexim1, illustrating specificity. It is possible that mir-16-1 microRNA enriched by the Hexim1 pull-down in Li et al. *NAR* 2007 is small and thus compatible with the 6-7 $\mu\text{m}^2/\text{s}$ fast-diffusing peak observed in Graham et al. *Mol Cell* 2025. These interesting observations further highlight the significance of our current study where we dissected Hexim–RNA binding specificity.

Figure citations - The text jumps back and forth between figures some in the wrong order and some figure citations seem to be entirely wrong e.g. Fig. 2B should be cited on p.5 (“...referred to as BR-L-AR in the following text.”) before Fig. 2A is coming in. There is no Fig.3E but it is referred to in the text (p.7 “...bulge (U40U41) is also present 3' to the alternative GAUC palindrome (Fig. 3E).”)

Response:

We have fixed these errors in the revised manuscript.

Chemical exchange - If this paper is meant for a general audience, the authors should explain what slow, intermediate and fast exchange mean in context of the system studied and the complexes that are formed here (e.g. on p.6, p.8).

Response:

Due to the manuscript length constraint, we have modified relevant text to give a brief explanation and cited a general review article for this purpose.

Abstract final sentence “These results provide mechanistic insights into Hexim–RNA specificity and explain how P-TEFb can be effectively regulated to respond to changing levels of transcriptional signaling.” - This conclusion is not warranted given that the authors have not carried out any assays in cells and used only artificial in vitro systems with few (and mostly severely truncated) components.

Response:

We have now separated the conclusion sentence and the broader implication sentence into two stand-alone sentences to avoid this confusion.

Figure inconsistencies - I understand that some peaks are broadened upon addition of RNA but it still looks like some bars are missing in Suppl.Fig.4C basic region in comparison to Fig. 2D/E? Also the baseline at 0 is missing in Suppl.Fig.4C.

Response:

Supplementary Fig 4C is secondary structural propensity scores, calculated from C-alpha and C-beta chemical shifts (see Methods for detail). The two residues that we could assign amide chemical shifts in Fig. 2C, however, could not be assigned both C-alpha and C-beta chemical shifts for the SSP calculation. We used carbonyl assignments to verify the amide resonances. Clarification of these and a line of zero SSP value have been added to Supplementary Fig. 5C and caption.

Suppl. Fig. 1 and 2: The sequence alignments seem to be cut out of larger alignments because they display numerous empty columns that hold no information. These empty columns should be removed to make the alignments more compact and easier to understand.

Response:

Yes indeed, we performed sequence alignments with 294 genes for Hexim1 and 210 genes for Hexim2 from NCBI gene cards, and randomly selected three genes each, in addition to human and mouse, for display (5 out of 294 for Hexim1 and 5 out of 210 for Hexim2). This information was described in Methods section titled “Multiple sequence alignment” for the details of alignment analysis. Many of the empty columns reflect differences between Hexim1 and Hexim2 paralogues. Other empty columns are important for us to understand which regions have insertions from orthologs, which also inform on whether adjacent structural elements are more separate modules or belong to a continuous domain. We have manually removed the empty columns resulting from full alignment insertions from the five representative gene displays, which mainly shrunk Supplementary Fig. 1, not much for Supplementary Fig. 2. We have added these now missing details of the insertions to Supplementary Fig. 1 caption.

REVIEWER COMMENTS

Reviewer #1 (Remarks to the Author):

The manuscript has been significantly improved, although it remains rather complicated to read in places. However, this is probably unavoidable when it comes to the description of NMR chemical shift perturbation data and the explanation of MTSL spin labelling experiments. The use of RNA constructs is a bit better to follow now and the confusion with mixed constructs in neighbouring panels is partly resolved. The discussion benefits a lot from the new model figure (Fig. 8) of the proposed regulation of P-TEFb by the 7SK snRNP in the main text. The figures are well prepared, although they may be difficult to follow for non-NMR specialists. I have no further comments to this expert study.

Response: We thank the reviewer for the positive feedback and for acknowledging our efforts to improve readability for non-NMR specialists. During the final revision, we have further refined the NMR sections to improve flow and readability, while preserving technical precision:

(1) We have modified Fig. 3 and Fig. 4 to specifically indicate the Pro residues and overlapping residues, and Fig. 6C by adding domain diagram.

(2) We revised the NMR-dense paragraphs lightly to improve the flow while maintaining the same technical details (Lines 142-161 and Lines 187-219).

Reviewer #3 (Remarks to the Author):

A few points to consider revising:

Review:

- Constructs were clarified reasonable, paper still hard to read

Response: We appreciate the reviewer's feedback. Please see response for Reviewer #1 regarding further edits we made make the paper easier to read.

- Fig. 1E: Where do percentages come from? How are the actual Mass-Photometry size correlated to the thought to be observed species... not clear

Response: The percentages are derived from the Mass Photometry analysis, which calculates the proportion of total particle counts represented by the Gaussian-fitted peaks. For instance, ~71% of total particle counts fall under the fitted peak corresponding to 49 ± 10.2 kDa for the BR-L-AR construct, while ~29% correspond to larger particles in the 60–160 kDa range. These observations are consistent with the presence of higher-order oligomers detected by native PAGE. This detail about percentages has been incorporated into Methods and Fig. S4 caption.

To further clarify this, we have revised the figure caption for Fig. 1E as follows:

“The observed Mass Photometry peak corresponds to a molecular weight of 49 ± 10.2 kDa—between the theoretical trimer (39.6 kDa) and tetramer (52.8 kDa) sizes, but closer to the latter. This shift likely results from the trimer peak being near the 30 kDa detection limit (gray arrow), causing partial peak truncation and overlap with the tetramer peak. Monomer and dimer species fall below the detection threshold and are not observed.”

- According to native gel (Fig. S4a), Hexim1 can also form tetramers. not indicated, please clarify

Response: We have now mentioned the minor tetrameric species of full-length Hexim1 observed by native PAGE at 4 μM protein concentration in the Fig. S4A caption (6.7% tetramer species by counting gel band intensities). This contrasts with the L-AR-CC construct, which lacks the BR and therefore does not form tetramers at 8 μM . At 40nM dilute condition, no tetramer species of Hexim1 were detected by Mass Photometry (Fig. S4E). The dimer bands being the major bands in native PAGE is now clarified in the main text (line 117).

- The other species in the mass photometry studies are ignored? Why, please explain
- o Fig. S4c/1E the calculation of trimer and tetramer are not evident
- o Which of the data for BR-L-AR is for 40 nM or 4-25 μM ?

Response: All Mass Photometry data were collected at 40 nM protein concentration. The concentrations and the higher-order oligomeric species (could not be fit well to gaussian functions due to their lower counts at 40nM and apparent heterogeneity) are now labeled on the Mass Photometry panels (Fig. 1E and Fig. S4). Details regarding BR-L-AR trimer and tetramer species are provided two points above.

- CSP mean doesn't make sense – one usually establishes CSP baseline that is not changing, and anything bigger than the std of this is considered a change in CSP... so with mean you are already excluding some relevant CSPs.

Response: One or two standard deviations above the mean CSP value across the entire protein is a widely accepted threshold for identifying residues with significant perturbations. This approach is appropriate here, given that several BR residues broaden beyond detection upon RNA binding—likely leading to underestimation of the largest CSP values.

We agree that residues showing weaker or transient interactions may fall below the mean+STD cutoff. To capture these subtler effects, we plotted both mean and mean+STD thresholds for protein CSPs (Fig. 2E) and three-tier thresholds (mean, mean+0.5 \times STD, and mean+STD) for RNA CSPs (Fig. 1B, Fig. 5). For instance, the PYNT motif (residues 202–209) displays CSPs between mean and mean+STD, consistent with its proposed transient intermolecular interactions with fast exchange dynamics in our study. That said, we chose to stay conservative with the main text description, due to the likely underestimation of the largest CSP.